



# Particulate organic nitrates in eastern China: variation characteristics and effects of anthropogenic activities

Jun Zhang[1], Xinfeng Wang[1], Rui Li[2], Shuwei Dong[1], Yingnan Zhang[1], Penggang Zheng[1], Min Li[1], Tianshu Chen[1], Yuhong Liu[1], Likun Xue[1], Wei Nie[3], Aijun Ding[3], Mingjin Tang[2], Xuehua Zhou[1], Lin Du[1], Qingzhu Zhang[1], Wenxing Wang[1]

[1]Environment Research Institute, Shandong University, Qingdao, 266237, China
[2]State Key Laboratory of Organic Geochemistry, Guangzhou Institute of Geochemistry, Chinese Academy of Sciences, Guangzhou 510640, China
[3]Joint International Research Laboratory of Atmospheric and Earth System Sciences, School of Atmospheric Sciences, Nanjing University, Nanjing, 210023, China

Correspondence: Xinfeng Wang (xinfengwang@sdu.edu.cn) and Wenxing Wang (wxwang@sdu.edu.cn)

**Abstract.** Particulate organic nitrates (PONs) constitute a substantial fraction of secondary organic aerosols and have important effects on the reactive nitrogen budget and air quality. Laboratory studies have revealed the non-negligible influence of the interactions between anthropogenic pollutants and biogenic volatile organic compounds (BVOCs) on the formation of PONs. In this study, the contents of specific PONs, including monoterpene hydroxyl nitrate (MHN215), pinene keto nitrate (PKN229), limonene di-keto nitrate (LDKN247), oleic acid keto nitrate (OAKN359), oleic acid hydroxyl nitrate (OAHN361), and pinene sulfate organic nitrate (PSON295), in fine particulate matters at four rural and urban sites in eastern China were determined, and the variation characteristics of PONs and the impacts of human activities on PONs formation were investigated. The average concentration of PONs ranged from 116 to 548 ng m$^{-3}$ at these four sites. PONs were present in higher levels in summer than in other seasons, owing to the high emissions of precursors and intensive photochemical activities in this hottest season. Among the six species of PONs, MHN215 was dominant. In addition, the proportion of OAKN359 in PONs in urban areas was much higher than that in the rural site, indicating that OAKN359 primarily originated from anthropogenic activities. Slight diurnal differences existed in the concentration and secondary formation of specific PONs and varied with locations, seasons, and precursor VOCs. The measurement results showed that PONs in North China were clearly influenced by coal combustion and biomass burning, while meteorological conditions and biogenic emissions were the dominant contributing factors in the South China. Biomass burning significantly enhanced the formation of PONs due to the elevated concentrations of ozone and the released BVOCs. Sulfur dioxide (SO$_2$) emitted from coal combustion was able to react rapidly with Criegee intermediates, the reaction products of BVOCs with ozone, to produce PONs at high rates, suggesting that there is a substantially greater role played by SO$_2$ in organic nitrate chemistry than has previously been assumed.

## 1 Introduction

Particulate organic nitrates (PONs) are important components of secondary organic aerosols (SOA) and affect the regional reactive nitrogen budget, tropospheric ozone (O$_3$) formation, and cloud condensation nuclei (CCN) activation. Field measurements have revealed that the total particle-phase organic nitrates contribute 2–12% of the total organic aerosol in the southeastern United States in summer (Rollins et al., 2012; Xu et al., 2015b). In addition, PONs serve as a sink of nitrogen oxides (NO$_x$) and can release NO$_x$ into the atmosphere in remote areas (Horowitz et al., 2007). As they possess a polar nitrate group, PONs can also change the aerosol hygroscopicity and thus alter the CCN activity of aerosols (Renbaum and Smith, 2009). The origins and formation processes of PONs have attracted close attention in the past decade, because of their important role in atmospheric chemistry.





Laboratory studies have revealed the potential precursors of PONs and proposed several PON formation mechanisms that have been applied in atmospheric models. The biogenic volatile organic compounds (BVOCs) isoprene, monoterpenes, sesquiterpene, and oleic acid have been identified as major precursors of PONs and the production yields are dramatically different with BVOC species (Docherty and Ziemann, 2006; Fry et al., 2014; Boyd et al., 2015). The possible formation mechanisms of PONs in both gas and aqueous phases have been investigated in laboratory studies. In the gas phase, the

oxidation of BVOCs induced by OH radicals or $O_3$ mainly takes place during the daytime, while $NO_3$ radicals are responsible for the production of organic nitrates at nighttime (Rollins et al., 2012; Ng et al., 2017). In the aqueous phase, oxidants (such as $O_3$ and radicals) react via electrophilic addition to double bonds or electron transfer with dissolved PON precursors (i.e., BVOCs) (Herrmann et al., 1999; Witkowski et al., 2018). Once formed, highly functionalized nitrates are expected to partition to the particle phase due to their higher vapor pressure (Lim and Ziemann, 2005). Moreover, laboratory

studies have shown that organic nitrates can be formed heterogeneously by a mechanism catalyzed by aerosol acidity (Surratt et al., 2008; Han et al., 2016). Nevertheless, field investigation by Xu et al. (2015a) has elucidated that the impact of sulfate on isoprene-derived SOA formation was direct (possibly through nucleophilic addition) rather than indirect (by liquid water or acidic species). Although BVOCs have been recognized as the major precursor of PONs and some formation processes have been proposed, the concentrations of PONs are usually underestimated in models, implying that current

knowledge of sources and mechanisms of formation of PONs is incomplete (Pye et al., 2015). Thus, there is an urgent need to explore the formation processes of specific organic nitrates in the complex, real atmosphere via direct field measurements.

    The abundance of PONs in ambient atmosphere varies with location, and their formation is apparently influenced by anthropogenic activities. It is of particular importance to unambiguously determine whether anthropogenic emissions significantly enhance the formation of PONs, to form a scientific basis for pollution control strategies. In Europe and

America, high concentrations of PONs (0.8–2.8 μg m$^{-3}$ in total) were usually observed at the sampling sites with large anthropogenic emissions, i.e., urban and suburban environments, whereas the abundance of PONs was generally low or below detection limits at remote and high-altitude sites (Day et al., 2010; Xu et al., 2015b; Kiendler-Scharr et al., 2016; Reyes-Villegas et al., 2018). Day et al. (2010) found that mixed urban fossil fuel combustion was primarily responsible for the high concentrations of PONs in southern California. The latest observations of PONs in China have shown that a large

disparity exists between the measurement methods and the concentrations of detected species, with the concentrations of PONs ranging from below detection limits to several micrograms per cubic meter found that mixed urban fossil fuel combustions were primarily responsible for the high concentrations of PONs in southern California. On the basis of latest observations of PONs in China, large disparity exist and the concentrations of PONs range from below detection limit to several micrograms per cubic meter (He et al., 2014; Ma et al., 2014; Zhang et al., 2016; Li et al., 2018; Wang et al., 2018;

Ye et al., 2019). Wang et al. (2018) suggested that future reduction in anthropogenic emissions would help to reduce the biogenic SOA loading because the abundances of organosulfates and nitrooxy-organosulfates increased significantly in acidic sulfate aerosols under high humidity. Moreover, Zhang et al. (2016) observed high concentrations of PONs in Beijing, China, of up to 4.2 μg m$^{-3}$ in total during the biomass burning period, indicating that burning sources are important contributors to PONs. It has been highlighted that the transformation of BVOCs to PONs can be greatly promoted in the

presence of substantial anthropogenic emissions such as $NO_x$ (Ng et al., 2007). However, specific PONs have different concentration trends and formation mechanisms, and their levels have not been meticulously measured in polluted areas and their proposed mechanisms of formation have not been fully understood. Therefore, a comprehensive evaluation of PONs via field observations in multiple locations in polluted regions in China is necessary.

    In this study, six major components of PONs were determined in fine particulate matter ($PM_{2.5}$) collected at both rural and

urban sites in polluted areas in eastern China. The concentration levels and the spatial and temporal variation characteristics are presented. The results confirm the strong influences of anthropogenic activities on the abundances and the formation of





PONs. The enhancements of PONs caused by biomass burning and coal combustion as well as the mechanism underpinning the formation of these compounds are then analyzed in details.

## 2 Experiments and methods

### 2.1 Site description

The sampling of $PM_{2.5}$ filter samples and the online measurements of related pollutants were conducted at four sites in eastern China, including a rural site in Dongying (DY) and three urban sites in Jinan (JN), Nanjing (NJ), and Guangzhou (GZ). The locations of these four sites can be seen in Fig. 1 and Table 1. Related sampling site information is described below.

The filter sampling and online measurements were conducted in both summer and winter at the rural site in DY. The DY site is situated at the Yellow River Delta Ecology Research Station of Coastal Wetland, Chinese Academy of Science, beside the Yellow River Delta National Nature Reserve, in Dongying, Shandong Province. The surrounding terrain is flat and surrounded by extensive wetlands and abundant biological resources, so this site is largely influenced by biogenic emissions. It is far from urban areas and local anthropogenic emissions are rare. More information about the DY site and the instruments deployed can be found in Zhang et al. (2019).

The JN site is located at the Atmospheric Environment Observatory Station in the Central Campus of Shandong University in Jinan, Shandong Province. Jinan City is one of the industrialized regions in North China with a large population and heavy traffic. A dozen heavy industries such as power plants, steel plants, cement plants, and chemical plants, are situated in the urban and suburban areas within 5–20 km of the sampling site. The shallow saucer-shaped terrain of the city aggravates air pollution by creating only weak atmospheric diffusion. Details about the JN site and the instruments deployed were given by Li et al. (2018).

The NJ site is situated in the SORPES station in Xianlin Campus of Nanjing University in Nanjing, Jiangsu Province. Located in the middle part of the downstream Yangtze River, Nanjing is a megacity dominated by financial and software industries. This site is surrounded by education and software technology districts. There are two busy roads near the site, i.e., the G25 Freeway 300 m to the west and the G312 National Road 1.8 km to the north, which possibly affect the air quality at the sampling site. Detailed descriptions of this station were given by Sun et al. (2018).

The southernmost GZ site is located at the Guangzhou Institute of Geochemistry, Chinese Academy of Sciences in Guangzhou, Guangdong Province. It is surrounded by education and residential districts, with two busy expressways nearby. There is high rainfall and abundant evergreen broad-leaved forest in Guangzhou due to the area's subtropical climate. The dominant vegetation types in this area are shrubs and tropical trees with high vegetation coverage, which emit a large quantity of BVOCs (Tang Hui, 2016). Detailed information of the GZ site can be found in Bi et al. (2016).

### 2.2 Sampling and measurements

$PM_{2.5}$ samples were collected by medium-volume $PM_{2.5}$ samplers (TH-150A, Wuhan Tianhong, China) at a flow rate of 100 L $min^{-1}$. Fine particulate matter was collected on quartz fiber filters that were prebaked prior to collection to remove any absorbed organic matter. The sampling periods at the four sampling sites are also contained in Table 1. Daytime (8:00–19:30, Beijing time, the same hereinafter) and nighttime (20:00–7:30) samples were collected separately. Before and after sampling, the filters were weighted by microbalance (ME5, Sartorius, Germany) at constant temperature and humidity to obtain the mass concentration. The filter samples were stored at −20 °C for subsequent chemical analyses.

Related trace gases and meteorological parameters were measured in real time at the DY site. The species measured were $NO_2$ (Teledyne Advanced Pollution Instrumentation, API, Model T500U, USA), CO (Teledyne API, T300U, USA), $O_3$ (Model 49C, Thermo Electronic Corporation, TEC, USA), and $SO_2$ (Model 43C, TEC). The meteorological parameters of temperature (T), relative humidity (RH), wind direction and speed were also monitored online (JZYG, PC-4, China). In



addition, VOCs samples were collected in 2 L stainless-steel canisters and were analyzed at the University of California at Irvine after the field campaigns. The water-soluble ions including potassium, sulfate, nitrate, and ammonium were analyzed by ion chromatography (Dionex Corporation, IC90s, USA) (Luo et al., 2020). For the JN, NJ and GZ sites, the concentrations of
trace gases in the nearest air quality monitoring stations were used for auxiliary analysis and the data were adopted from the China National Environmental Monitoring Centre (http://106.37.208.233:20035/). The meteorological data for these three sites were taken from the Weather Underground (https://www.wunderground.com).

### 2.3 Chemical analysis

To determine the content of organic carbon (OC) and elemental carbon (EC) in fine particulate matters, a punch (2 cm$^2$) of
each filter sample was analyzed by a thermal-optical transmittance method according to the protocol of NIOSH (Sunset, OCEC analyzer, USA). The minimum OC/EC ratio method (Turpin and Huntzicker, 1995) was used to estimate the concentrations of secondary organic carbon (SOC) at each site. Furthermore, according to the ratios of SOA to SOC (Turpin and Lim, 2001; Liu et al., 2017), the SOA concentrations were calculated from the SOC concentrations by multiplying by a factor to give an estimate, i.e. 2.1, 2.2, 1.8, 1.6 and 1.6 for DY site in winter, DY site in summer, GZ, NJ, and JN sites,
respectively.

Detailed treatment procedures and analytical methods for organic nitrates used in this study can be found in Li et al. (2018). However, to preclude the influence of OH and methyl radicals that are probably produced during ultrasonic extraction (Miljevic et al., 2014), we used a thermostatic orbital shaker as a substitute of ultrasonication during the extraction method. Briefly, the sample filters were extracted with methanol and concentrated by a rotary evaporator. The residue was
passed through 0.45 µm porosity PTFE syringe filters, and the resulting clear filtrate was taken to dryness under a gentle nitrogen gas stream at ambient temperature. The residues were immediately reconstituted with 400 µL of 2 mg L$^{-1}$ tropinone (> 98%, IL, USA) in methanol, which was used as the internal standard to minimize instrumental errors. In addition, the surrogate standard was added to blank filters, and these were subsequently treated and analyzed in the same way as the sample filters, to evaluate the recovery of the analytical method.

Sample extracts were analyzed by ultra-performance liquid chromatography (Accela 1250, Thermo Scientific, USA) coupled with electrospray ionization and triple quadrupole tandem mass spectrometry (TSQ Vantage, with a FWHM resolution of 7500, Thermo Scientific) and ultra-high performance liquid chromatography (Ultimate 3000, Thermo Scientific) equipped with electrospray ionization and Orbitrap mass spectrometry (Q Exactive Focus, with a FWHM resolution of 70000, Thermo Scientific) in both positive and negative ion modes. Chromatographic separations were performed with an
Atlantis C18 column (2.1×150 mm, 2.1 µm) which was operated at 24 °C. The mobile eluents were (A) methanol (HPLC-grade, Sigma-Aldrich, USA) and (B) 0.1% formic acid (HPLC-grade, Sigma-Aldrich, in water (v/v)) at a total flow rate of 0.2 mL min$^{-1}$. When the sample extracts were detected in positive mode, the gradient elution procedure was set as follows: the composition started with 30% A and 70% B and held for 3 min; increased to 90% A within 10 min and then held for 3 min; decreased back to 30% A within 0.1 min and then held for 3.9 min. As for the negative mode, the gradient elution
program was as follows: eluent A initially was set at 3% for 3 min; increased to 90% within 12 min and held for 7 min; decreased to 3% within 5 min and held for 8 min. The mass spectrometers were operated with the capillary voltage at 4.0 kV and the capillary temperature at 320 ℃.

Six kinds of organic nitrates were identified from the methanol extracts of PM$_{2.5}$ samples (as listed in Table 2). By monitoring the [M+H]$^+$ ions in positive ion mode, monoterpene hydroxyl nitrate (MW = 215, MHN215) and pinene keto
nitrate (MW = 229, PKN229) were recognized from the neutral losses of fragments of 18 Da (H$_2$O) and 46 Da (NO$_2$) by MS$^2$ and MS$^3$ spectrum which were consistent with the structure of analytes of one hydroxyl group and one nitrooxy group (Perraud et al., 2010). Limonene di-keto nitrate (MW = 247, LDKN247) were confirmed with the losses of fragments of 17 Da (OH), 18 Da (H$_2$O), 49 Da (OH+H$_2$O), and 46 Da (NO$_2$) which were consistent with the LNKN247 structure of two





carbonyl groups, one hydroxyl group and one nitrooxy group. Oleic acid keto nitrate (MW = 359, OAKN359) was identified

with the losses of fragments of 18 Da ($H_2O$), 63 Da ($HNO_3$), and 92 Da ($HNO_3+C_2H_5$) which represented the carbonyl and the nitrooxy groups of OAKN359 (Docherty and Ziemann, 2006). Oleic acid hydroxyl nitrate (MW = 361, OAHN361) was identified the losses of fragments of 18 Da ($H_2O$), 63 Da ($HNO_3$), and 106 Da ($NO_3+CO_2$) which were consistent with the hydroxyl, the nitrooxy and the carboxyl groups. More detailed information can be seen in our previous study by Li et al. (2018). Via monitoring the $[M-H]^-$ ions in negative ion mode, pinene sulfate organic nitrate (MW = 295, PSON295) was

identified based on the neutral losses of fragments of 47 Da ($HNO_2$), 63 Da ($HNO_3$) and 74 Da ($CO+NO_2$) and ion losses of 96 Da ($SO_4^-$) and 80 Da ($SO_3^-$) by $MS^2$ and by $MS^3$ spectrum, respectively (Surratt et al., 2008; He et al., 2014). The estimated molecular formulas of the analytes were calculated within a mass tolerance of 4 ppm. Note that the nitro-aromatic hydrocarbons and nitro phenolic compounds in particulate matters are believed to have little interference on the identifications of organic nitrates due to the different molecular weights and the common detection mode of negative ion.

Due to the lack of commercial standards, .MHN215, PKN229, and LDKN247 were quantified with the surrogate standard of ($1R$, $2R$, $5R$)–(+)-2-hydroxy-3-pinanone (GC-grade, TCI, Japan) with consideration of the similar retention time and structures or functional groups. OAKN359 and OAHN361 were quantified with the surrogate standard of ricinoleic acid (GC-grade, TCI) due to the similar structures. Furthermore, PSON295 was quantified with the surrogate standard of (-)-10-camphorsulfonic acid (GC-grade, TCI) (Worton et al., 2011; He et al., 2014). Although the selected three surrogate

standards have no nitrooxy group, they are the most suitable commercial compounds among the potential surrogate standards that we had tried. Six-point linear curves of peak area (normalized by the internal standard) versus concentration were established ($r^2$ >0.99) for further quantification. For the molecule with several isomers, quantification was performed by summing up the total peak areas of the isomers which were treated as one kind of organic nitrate. The recoveries of the three surrogate standards were all approximately 60%. Note that the use of surrogate standards will enlarge the uncertainty in the

absolute concentrations of PONs, as the ESI response factor may vary for different compounds. In addition, the extracts of $PM_{2.5}$ samples were analyzed partly by the triple quadrupole tandem mass spectrometer and partly by the Orbitrap mass spectrometer. The determination results from both mass spectrometers for the same samples showed good consistency, indicating the reliability of the quantifications of organic nitrates from the low-resolution triple quadrupole tandem mass spectrometer.

### 3. Results and discussion

### 3.1 Spatial and seasonal variations of PONs

The measurement results at the four sampling sites in eastern China show that the concentrations of PONs were relatively high and exhibited distinct spatial and seasonal variations. Table 3 summarizes the average concentrations of specific PONs (MHN215, PKN229, LDKN247, OAKN359, OAKN361, and PSON295) and related pollutants and parameters at the DY,

GZ, NJ, and JN sites. As shown, the average total concentration of the six PONs (ΣPONs) during the measurement periods was in the range of 116–548 ng m$^{-3}$. Among these six organic nitrates, MHN215 was the dominant species, with an average concentration of 50.7–243 ng m$^{-3}$, followed by OAKN359 (38.5–194 ng m$^{-3}$), PKN229 (8.3–164 ng m$^{-3}$), LDKN247 (10.4–27.0 ng m$^{-3}$), PSON295 (3.6–8.7 ng m$^{-3}$), and OAHN361 (0.8–7.3 ng m$^{-3}$). During the observation periods, the most abundant PONs were observed at the DY and GZ sites in summer (with the average concentrations of ΣPONs of 548 and 512

ng m$^{-3}$, respectively), followed by the DY site in winter (241 ng m$^{-3}$) and the NJ site in autumn (196 ng m$^{-3}$), with the least at the JN site in autumn (116 ng m$^{-3}$). The maximum 12-h average concentration of ΣPONs was as high as 1248 ng m$^{-3}$, and occurred at the DY site on June 5, 2017. At these four sites, the contribution of PONs to $PM_{2.5}$ was generally small, with the average mass ratio of ΣPONs to $PM_{2.5}$ ranging from 2.6‰ to 5.7‰. Nevertheless, the PONs still accounted for a notable fraction of SOA, with the average mass ratio of ΣPONs to SOA varying from 2.1% to 14.7%. Note that though the data of





PONs at each site were limited, the observation results could generally represent the variation characteristics in different sites and seasons with examination of the concentrations of other air pollutants (e.g., $O_3$, $SO_2$, $NO_2$, OC, and $PM_{2.5}$) during, before and after the sampling periods.

A comparison of different sampling sites shows that elevated PONs concentrations usually appeared at the regions with high vegetation cover. Considering the plethora of trees and grasses surrounding the rural site of DY, we attributed the

enhanced formation of PONs (indicated by elevated ratios of $\Sigma$PONs/SOA) at the DY site primarily to the intense emissions of BVOCs. The similar situation can also be seen in urban Guangzhou. The higher levels of PONs and the higher ratios to $PM_{2.5}$ and SOA at GZ site than those at the other two urban sites of NJ and JN is primarily ascribed to the massive broad-leaf trees in the subtropical region. In addition, the concentration of PONs tended to increase in the hot season. The average abundance of PONs at the DY site in summer was almost twice of that in winter, indicating that the intense photochemical

activities (e.g. 71.0 ppb of ozone) and increased emission of BVOCs in summer facilitated the secondary formation of PONs.

In previous studies, the total PONs detected at Kleiner Feldberg mountain in summer (Sobanski et al., 2017), in Colorado in summer (Fry et al., 2013), in Manchester in winter (Reyes-Villegas et al., 2018), and at Noto peninsula from December 2012 to December 2015 (Sadanaga et al., 2019) were at the average level of 1–4 μg m$^{-3}$. The concentrations of individual PONs determined recently in Jinan and Beijing in spring varied from 13.8 to 325 ng m$^{-3}$ (Li et al., 2018; Wang et al., 2018),

which are generally comparable to concentrations observed at the four sites in eastern China in this study. The concentrations of PONs measured in this study are substantially lower than the previously reported values for total particulate organic nitrates, indicating that a large fraction of PONs have not been identified and quantified in this work and require investigations with advanced measuring techniques in the future.

In addition to the site-to-site variation in total PON abundance, there were also site-to-site variations in the fractions of

individual PONs within $\Sigma$PONs. As shown in Fig. 2, MHN215 constituted the largest fraction of the identified PONs at all four sampling sites, accounting for 35–58%, followed by OAKN359 and PKN229. The highest proportion of MHN215 (58%) in $\Sigma$PONs was observed at the DY site in summer, while the lowest proportion (35%) was found at the NJ site in autumn. The sum of monoterpene-derived organic nitrates (i.e. MHN215, OAKN229, and LDKN247) constituted the largest proportion of the observed PONs, with a total contribution of 60–82%. It is worth noting that OAKN359 also contributed to

a large fraction (16–38%) of the $\Sigma$PONs and it was more abundant at urban sites (33–38%) than at rural sites (16–17%). Additionally, PSON295 and OAHN361 only accounted for small proportions of the $\Sigma$PONs. It is interesting that the proportion of PSON295 (6%) at the JN sites was markedly higher than those at other sites (<3%).

The dominance of monoterpene-derived organic nitrates in PONs was also observed in southeastern United States (23–44% in mole) and in southwestern China (Ding et al., 2016) and was ascribed to the high emissions of monoterpenes together with

the relatively high production yields of PONs. In addition, there are more formation pathways for OAKN359 from oleic acid than for OAHN361 (Docherty and Ziemann, 2006) and the structure of the former is more stable in the oxidizing atmosphere. As a result, the concentrations of OAKN359 are much higher than those of OAHN361, especially in the areas with high $NO_x$ emissions. The large contribution of OAKN359 in urban areas indicates the strong impact of anthropogenic activities on the formation and the abundance of OAKN359 in ambient air. When used in cooking at high temperatures, oleic acid is

volatilized into the atmosphere, where it reacts with $O_3$ and $NO_2$ to produce organic aerosols including OAKN359 (Schauer et al., 2002). Oleic acid has been identified as an important component of urban aerosols and serves as an indicator of the contribution of meat cooking to particulate pollution (Rogge et al., 1991). The identification and quantification of oleic acid-derived organic nitrates in this study confirmed the significant influence of cooking and oil processing on the SOA in urban and rural areas in eastern China. As for PSON295, the relatively high contribution at urban Jinan is possibly associated

with the massive coal-fired industries distributed throughout the city.





### 3.2 Diurnal difference of PONs

Generally, the daytime concentrations of ΣPONs were comparable to those at nighttime (p > 0.1 for most sites). However, some differences were found between the daytime and nighttime concentrations of individual PONs and a disparity was noticed between the diurnal differences of monoterpenes- and oleic acid-derived PONs, indicating a dependence of diurnal

patterns on the precursor VOCs. For example, at the NJ site, monoterpene-derived PONs (i.e. MHN215, PKN229, and LDKN247) exhibited slightly higher concentrations at nighttime than daytime, while oleic acid-derived PONs (e.g. OAKN359) presented the opposite diurnal pattern (see Fig. 3). In addition, the diurnal difference varied from site to site. At the NJ and GZ sites, the monoterpene-derived PONs were slightly more abundant at nighttime than daytime. Nevertheless, they were to more abundant at daytime than nighttime at the DY site in summer and at the JN site in autumn. The different

diurnal patterns of monoterpene-derived PONs were believed to be associated with the various monoterpene emission characteristics and the diverse oxidation capacity at daytime and nighttime in different regions.

In addition, the production of PONs relative to SOA also exhibited apparent diurnal variation. Generally, there were more PONs produced at the sampling sites with higher vegetation coverage (i.e. the DY and GZ sites) and the diurnal differences in PONs production were more distinct at rural sites than at urban sites, according to the PONs/SOA ratios (see Fig. 4). At

the DY site in summer, more PONs were produced during daytime than nighttime, but in winter the situation was the opposite. At urban sites, the production of most specific PONs during daytime and nighttime were comparable, whereas a few individual PONs (e.g. MHN215 and PSON295) were formed in slightly higher concentrations at nighttime than daytime.

Our results indicate that the diurnal differences in abundance and formation of PONs were not only related to site locations but also associated with seasonal variations. At the rural site of DY in summer, high emissions of BVOCs

precursors and the strong atmospheric oxidation capacity during daytime enhanced the secondary formation processes compared to those at nighttime. The higher concentrations of some PONs at nighttime than daytime in winter is suggestive of the important role of $NO_3$ radicals and possibly aqueous reactions in the oxidation of BVOCs. With regard to the urban sites, we infer that anthropogenic pollutants (such as NO and alkenes) underwent competitive reactions with $NO_3$ radicals which thus decreased the nighttime production of PONs from BVOCs.

### 3.3 Influences of anthropogenic activities on PONs


The large variations in abundances and compositions of PONs among different sampling sites and seasons in eastern China were mainly caused by variations in precursors, oxidants, chemical processes, and atmospheric conditions. To understand the potential influences of anthropogenic activities on the observed PONs, the time series of PONs and the relationships with source tracers at difference sites are analyzed in detail here.

### 3.3.1 Effect of coal combustion


Figure 5 presents the temporal variations of concentrations of PONs, $SO_2$, CO, and $PM_{2.5}$ as well as the PONs/SOA ratio at the DY site in winter and at the JN site in autumn. Examination of the relevant tracer pollutants revealed that the abundance of PONs at the DY site in winter and at the JN site were obviously influenced by coal combustion. Based on the observation data of PONs, it can be seen that the concentration of PONs concurrently varied with the concentrations of $SO_2$, CO, and

$PM_{2.5}$ at the two sites. At the DY site in winter, the most abundant PONs (395 ng m$^{-3}$) level occurred during the daytime on 19 January, accompanied by highest levels of $SO_2$ and $PM_{2.5}$ (13.2 ppb and 229 μg m$^{-3}$, respectively). The elevated concentration of $SO_2$ on 19 January was attributed to the use coal combustion for heating in the rural areas surrounding the DY site in the cold season. Note that the ratio of PONs/SOA was not high on that day because of the offset effect caused by the increased SOA.

At the JN site, the proportion of PSON295 in SOA was noticeably higher than at the other sites. The concentrations of PONs in the daytime on 22 and 23 October (221 and 211 ng m$^{-3}$, respectively) were about twice as high than the average



value during the sampling period, with MHN215, LDKN247, and OAHN361 reaching their maximum concentrations during the JN observations. At the same time, the ratio of PONs/SOA was quite high, indicating the enhanced formation of PONs. The relatively high levels of $SO_2$ on 22 and 23 October were ascribed to the transport of plumes from coal-fired industries such as steel plants and thermal power plants in the northeast sector of Jinan City under the influence of northeast winds. Overall, the concurrent appearance of elevated concentrations of PONs and $SO_2$ at the DY site in winter and at the JN site was suggestive of the potential influence of coal combustion on the abundances of PONs.

Further correlation analyses showed that PONs and $SO_2$ at the DY site in winter and at the JN site exhibited strong relationships, with the Pearson's correlation coefficients being r = 0.89 and 0.85, respectively (p < 0.01, see Fig. 6(a)), and the ratio of PONs/SOA also correlated moderately with $SO_2$ at the JN site (r = 0.51, p < 0.05, not shown here). Among the six species of PONs, MHN215 and $SO_2$ showed remarkable correlations with r = 0.91 and 0.83 (p < 0.01, see Fig. 6 (b)) at the two sites. In addition, LDKN247 was moderately correlated with $SO_2$ at the DY site in winter (r = 0.57, p < 0.05), and this correlation was greater at the JN site (r = 0.80, p < 0.01, see Fig. 6 (c)). Nevertheless, the relationship between PONs and $NO_2$ was very weak (r < 0.3, not shown here), which indicates that vehicle exhaust had little influence on the abundances of PONs. The strong correlations between PONs and $SO_2$ at the DY site in winter and at the JN site confirmed the large effect of coal combustion on the formation of PONs during the measurement periods.

It is interesting and surprising that there were close relationships between PONs and $SO_2$ at the DY site in winter and at the JN site, because most PONs do not contain sulfur except for PSON295, and this species comprised only a small fraction of PONs. The chemical mechanism sheds light on the roles played by $SO_2$ in the formation regimes of PONs, which did not receive adequate attention in previous studies. The scheme of formation pathways of LDKN247 (see Fig. S1) provided by Master Chemical Mechanism version 3.3.1 (MCM, http://mcm.leeds.ac.uk/MCM/) shows that $SO_2$ directly reacts with the Criegee intermediates (CIs) derived from limonene and ozone. The reactions between $SO_2$ and CIs are proved to be quite fast by theoretical quantum mechanical studies, field studies, and laboratory experiments (Jiang et al., 2010; Berndt et al., 2012; Mauldin et al., 2012; Welz et al., 2012): on the order of $10^{-11}$–$10^{-13}$ $cm^3$ molecules$^{-1}$ s$^{-1}$, depending on the structure of the CIs. Based on the rate coefficient of $8 \times 10^{-13}$ $cm^3$ molecules$^{-1}$ s$^{-1}$ for CIs from the ozonolysis of limonene given by Mauldin et al. (2012) (not the rate coefficient of $1 \times 10^{-14}$ $cm^3$ molecules$^{-1}$ s$^{-1}$ in the MCM) and the calculation methods reported by He et al. (2014), the estimated average contributions of reactions involving $SO_2$ to the formation of LDKN247 at daytime and nighttime reached 9.3% and 6.3%, respectively. This indicated that the production of LDKN247 could be accelerated in the presence of high concentrations of $SO_2$ in ambient air.

More recently, the important contributions of monoterpene ozonolysis reactions to CIs formation and $SO_2$ oxidation have been revealed by chamber and theoretical studies (Newland et al., 2018). Therefore, we hypothesized that MHN215 can be formed from pinene-derived CIs, because its precursor, pinene, has a double bond on the ring which is in a chemically similar environment to that of the double bond in limonene. The above results imply that $SO_2$ plays a substantially greater role in the tropospheric chemistry of organic nitrates than had previously been assumed.

Furthermore, $SO_3$, one of the products of the reactions between $SO_2$ and CIs, and which is the anhydride of sulfuric acid, serves as an important nucleating agent and was conducive to SOA production. In sum, here we suggest that the high concentrations of $SO_2$ promoted the formation of PONs in two ways, i.e. by directly participating in the generation of PONs in the presence of ozone, and by indirectly facilitating the formation of SOA including PONs via providing seed aerosols (Chu et al., 2016; Zhao et al., 2018). At present, the reaction of unsaturated BVOCs (e.g. pinene) with ozone is recognized as one of the main pathways for SOA formation (Berkemeier et al., 2016). However, the subsequent reaction with $SO_2$ has not been fully considered in chemical models related to organic nitrates. This updated understanding of the formation pathways of PONs is of particular importance to generating a comprehensive evaluation of the sources and sinks of PONs.

In addition to high levels of $SO_2$, the increase in concentrations of PONs was always accompanied by increases in levels of inorganic salts (e.g., sulfates and nitrates), which were mostly produced from the gaseous precursors emitted from coal



330  combustion and other anthropogenic activities. According to the salting-in effect proposed by Xu et al. (2015a), rising salt
concentrations in aqueous solution would increase the solubility of polar organic compounds. It appears that inorganic salts
in fine particles also affected the formation of PONs through multiphase chemistry. Overall, our observation results and
deductions confirm that coal combustion significantly enhanced the formation of PONs in North China during the study
period and provides clear evidence for the active interactions between BVOCs and anthropogenic emissions.

### 3.3.2 Effect of biomass burning

Figure 7 shows the time series of concentrations of PONs, methyl chloride ($CH_3Cl$, a tracer of biomass burning), potassium
($K^+$, a tracer of biomass burning), ozone, the concentration product of $NO_2$ and ozone, and the PONs/SOA ratio during
daytime and nighttime at the DY site in summer. It was found that the abundances of PONs at the DY site in summer were
influenced by different factors at daytime and nighttime. During the daytime, the concentrations of PONs changed positively
340  with the concentration of ozone. Exposed to ozone-rich atmosphere, PONs concentrations were less correlated with $SO_2$
concentrations ($r = 0.35$, $p=0.36$) but strongly correlated with ambient ozone concentrations (Fig. 8, $r = 0.79$, $p < 0.05$).
During nighttime, PONs production was dominated by the source strength of $NO_3$ radicals, as Rollins et al. (2012) reported.
The correlation coefficients of PONs and PONs/SOA versus $NO_3$ radicals source strength (i.e. the concentration product of
$NO_2$ and ozone) was 0.89 ($p < 0.05$) and 0.62 ($p < 0.1$), respectively.

345  The concentration peak of PONs of 921 ng $m^{-3}$ was accompanied by the high level of ozone of 118 ppb in the daytime on
15 June. At the same time, it was noticed that the ratio of PONs/SOA and secondary inorganic ions of sulfate, nitrate, and
ammonium (not shown here) increased sharply, which indicated a large quantity of PONs had been produced. A significant
increase in the concentrations of hydrocarbons (in particular, aromatic compounds) was also observed, which favored ozone
formation. In addition, the concentrations of methyl chloride and potassium soared on 15 June (as shown in Fig.7),
accompanied by distinct and intensive fires that had newly appeared to the west of the sampling site. Further examination of
48-h backward trajectories using the HYSPLIT model (GDAS data, http://ready.arl.noaa.gov/archives.php) showed that the
air masses came from the west sectors of Shandong Province where the fires were located (see Fig. S2). Overall, biomass
burning facilitated ozone formation (Ma et al., 2018), and this further increased the production of PONs. Moreover, biomass
burning also led to increased emissions of BVOCs, as was indicated by the doubled concentrations of isoprene and pinenes
on that day, thus also contributing to PONs formation when exposed to high levels of ozone.

### 3.3.3 Effect of relative humidity and $NO_3$ radicals

At the GZ and the NJ sites, the concentrations of gaseous pollutants were generally unchanged during the sampling periods,
with no distinct pollution events evident. It was observed that the meteorological conditions had some influence on the
concentration of PONs. Fig. 9(a) shows the relationship between PONs and RH at daytime at the two sites. During the
daytime, PONs correlated with RH strongly at the GZ site ($r = 0.71$, $p < 0.05$) and moderately at the NJ site ($r = 0.56$, $p =$
0.12). PONs/SOA also correlated with RH ($r =0.41$, $p < 0.1$) at the GZ site, suggesting the potential promotional effect of
water on the formation of PONs. As the humidity increased, the fine particles were primarily in a liquid-like state during the
daytime (Slade et al., 2019), which affected the gas-aqueous partition and increased the heterogeneous uptake of precursors,
oxidants, and products (Wang et al., 2017). Aljawhary et al. (2016) has demonstrated that aqueous processes are an
additional source of α-pinene oxidation products. Therefore, we speculate that the high humidity (mostly above 60%) at the
GZ site accelerated the production of PONs aqueous reactions in cloud water or in aerosol liquid water.

During nighttime, the abundance of PONs at the NJ site was clearly influenced by $NO_3$ radicals, which was indicated by
the good correlation between PONs concentration and the concentration product of $NO_2$ and ozone (see Fig. 9(b)). Both the
linear slope and the correlation coefficient between these species at the NJ site ($k = 0.067$ and $r =0.78$) were lower than those
at the DY site in summer ($k = 1.33$ and $r = 0.89$, shown in Fig. 8) but higher than those at the JN site ($k = 0.016$ and $r = 0.25$,





not shown here), indicating there was a different production efficiency of these species in different locations. The production efficiency of PONs at nighttime by $NO_3$ radicals was substantially higher at the rural site than at the urban sites, probably due to the competing reactions of $NO_3$ radicals with NO and alkenes in urban areas. The very weak correlation of PONs with the concentration product of $NO_2$ and ozone at the GZ site was attributed to intensive emission of NO from busy traffic on

the expressways nearby.

**4 Conclusion**

Six kinds of PONs at four rural and urban sites in eastern China were determined and the pollution characteristics as well as the major influencing factors of these species were investigated. Elevated concentrations of PONs were observed in summer, and MHN215 was the dominant species. The diurnal patterns of PONs differed with the sampling sites and the precursor

VOCs, with similar day-night variations among MHN215, PKN229, and LNKN247, while OAKN359 and OKHN361 exhibited the opposite pattern. It was found that coal combustion, biomass burning, and ambient humidity had strong influence on the abundance and the formation of PONs. Temporal variations and correlation analysis showed that concentrations of PONs in the north of eastern China were clearly influenced by anthropogenic activities (i.e. coal combustion and biomass burning), while meteorological conditions (in particular the humidity) were key influencing factors

in the south. It is crucial to recognize that $SO_2$ from coal combustion plumes readily reacted with BVOCs-derived CIs, and that the resulting products transformed into PONs at high rates, indicating that there is a substantially greater role played by $SO_2$ in organic nitrate chemistry than has been previously assumed. Biomass burning strongly promoted the formation of ozone and emissions of BVOCs, and thus facilitated the production of PONs. This study highlights the significant effects of anthropogenic emissions on the concentration and production of PONs in fine particulate matter via interactions between

anthropogenic pollutants and BVOCs.

*Data availability.* The data presented in this article are available from the authors upon request (xinfengwang@sdu.edu.cn).

*Information about the Supplement.* In this supplement are provided the formation mechanism of LDKN247 from limonene as provided by MCM software, and the distribution of fires and 48 h backward trajectories on 15[th] June, 2017 at the DY site.

*Author contributions.* JZ conducted sample analysis and data analysis and drafted the initial manuscript. XFW funded the

deployment, directed the experiments, and revised the manuscript. RL developed the detection method and collected some samples. SWD performed the analysis of OC and EC. YNZ provided the information of MCM. Other authors contributed to providing sampling sites, providing instruments, providing measurement data, sample collection, sample treatment, and data acquisition, and commented on the manuscript.

*Competing interests.* The authors declare that they have no conflict of interest.

*Acknowledgements.* This work was supported by National Natural Science Foundation of China (nos. 41775118, 91644214, 91544213, 41675118), the National Key Research and Development Program of China (no. 2016YFC0200500), and the State Key Laboratory of Organic Geochemistry, GIGCAS (Grant No. SKLOG-201616). The authors would like to thank the website of China National Environmental Monitoring Centre (http://106.37.208.233:20035/) and Weather Underground (https://www.wunderground.com) for providing the gas and meteorological data.





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





**Tables:**

**Table 1. Locations of the sampling sites and the sampling periods.**

| Sampling site | Type | Location | Sampling period | Season |
|---|---|---|---|---|
| Dongying | Rural | 37°76′ N, 118°98′ E | 2017/1/15-2017/1/23<br>2017/6/4-2017/7/9 | Winter<br>Summer |
| Jinan | Urban | 36°40′ N, 117°03′ E | 2017/10/20-2017/11/1 | Autumn |
| Nanjing | Urban | 32°12′ N, 118°97′ E | 2017/10/19-2017/11/1 | Autumn |
| Guangzhou | Urban | 23°15′ N, 113°37′ E | 2017/6/24-2017/7/8 | Summer |

**Table 2. Formulas and possible structures of the identified PONs.**

| Particulate organic nitrates | Formula | Structure/potential structure | Reference |
|---|---|---|---|
| Monoterpene hydroxyl nitrate (MW = 215, MHN215) | $C_{10}H_{17}NO_4$ | | Fry et al., 2009; Perraud et al., 2010; Fry et al., 2011; Li et al., 2018 |
| Pinene keto nitrate (MW = 229, PKN229) | $C_{10}H_{15}NO_5$ | | Fry et al., 2011 |
| Limonene di-keto nitrate (MW = 247, LDKN247) | $C_{10}H_{17}NO_6$ | | Fry et al., 2011 |
| Oleic acid keto nitrate (MW = 359, OAKN359) | $C_{18}H_{33}NO_6$ | | Docherty and Ziemann, 2006 |
| Oleic acid hydroxyl nitrate (MW = 361, OAHN361) | $C_{18}H_{35}NO_6$ | | Docherty and Ziemann, 2006 |
| Pinene sulfate organic nitrate (MW = 295, PSON295 | $C_{10}H_{17}NO_7S$ | | Iinuma et al., 2007; Surratt et al., 2008 |




**Table 3. Statistics of concentrations of PONs and related pollutants and parameters at the four sampling sites (average ± standard deviation).**

| Species/ parameter | Dongying Winter | Dongying Summer | Guangzhou Summer | Nanjing Autumn | Jinan Autumn |
|---|---|---|---|---|---|
| MHN215 (ng m$^{-3}$) | 112 ±44.8 | 229 ±193 | 243 ±76.7 | 68.0 ±24.9 | 50.7 ±35.7 |
| PKN229 (ng m$^{-3}$) | 62.5 ±22.0 | 164 ±142 | 40.3 ±35.2 | 45.4 ±24.0 | 8.3 ±3.8 |
| LDKN247 (ng m$^{-3}$) | 18.2 ±5.94 | 26.0 ±23.3 | 27.0 ±9.01 | 13.9 ±7.46 | 10.4 ±8.59 |
| OAKN359 (ng m$^{-3}$) | 41.0 ±14.3 | 132 ±62.9 | 194 ±54.7 | 63.9 ±49.2 | 38.5 ±17.9 |
| OAHN361 (ng m$^{-3}$) | 7.3 ±10 | 1.6 ±1.2 | 0.9 ±0.8 | 0.9 ±0.4 | 0.8 ±0.9 |
| PSON295 (ng m$^{-3}$) | – | 8.7 ±15 | 7.7 ±7.6 | 3.6 ±2.9 | 7.1 ±7.7 |
| ΣPONs (ng m$^{-3}$) | 241 ±77.3 | 548 ±300 | 512 ±100 | 196 ±67.1 | 116 ±60.3 |
| NO$_2$ (ppb) | 7.6 ±8.4 | 4.8 ±2.8 | 20.0 ±3.93 | 30.9 ±11.1 | 34.6 ±16.3 |
| O$_3$ (ppb) | 17.2 ±9.71 | 71.0 ±26.6 | 13.0 ±13.5 | 23.0 ±16.2 | 20.7 ±14.7 |
| SO$_2$ (ppb) | 3.9 ±3.1 | 3.0 ±1.6 | 3.0 ±0.52 | 3.3 ±0.78 | 4.2 ±1.9 |
| CO (ppm) | – | 0.4 ±0.2 | 0.6 ±0.1 | 0.5 ±0.1 | 0.8 ±0.4 |
| OC (μg m$^{-3}$) | 7.1 ±5.7 | 8.0 ±4.0 | 5.4 ±1.5 | 9.3 ±3.0 | 11 ±5.6 |
| PM$_{2.5}$ (μg m$^{-3}$) | 76 ±48 | 106 ±45 | 78 ±7 | 55 ±24 | 64 ±40 |
| ΣPONs/PM$_{2.5}$ (‰) | 3.7 ±1.6 | 5.7 ±4.2 | 5.0 ±1.5 | 4.1 ±2.1 | 2.6 ±2.3 |
| ΣPONs/SOA (%) | 12.3 ±8.68 | 14.7 ±15.4 | 17.4 ±9.31 | 3.1 ±2.0 | 2.1 ±1.2 |
| Temperature (℃) | -1.4 ±1.7 | 26.0 ±4.03 | 27.1 ±3.12 | 14.3 ±3.32 | 12.2 ±3.37 |
| RH (%) | 53.5 ±11.9 | 61.2 ±13.1 | 79 ±11.5 | 67.0 ±15.5 | 45.9 ±13.2 |





**Figures:**

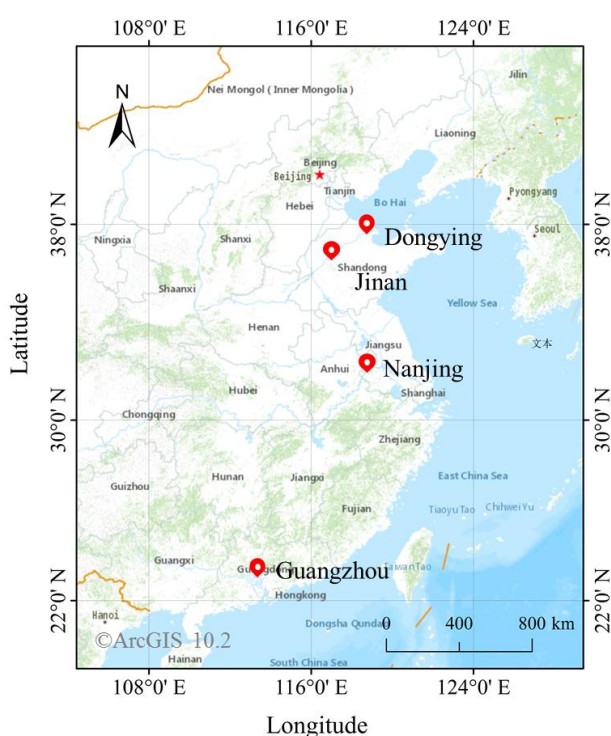

**Figure 1. Locations of the four sampling sites involved in this study. The map was provided and created by the software ArcGIS (Version10.2, ESRI, USA).**

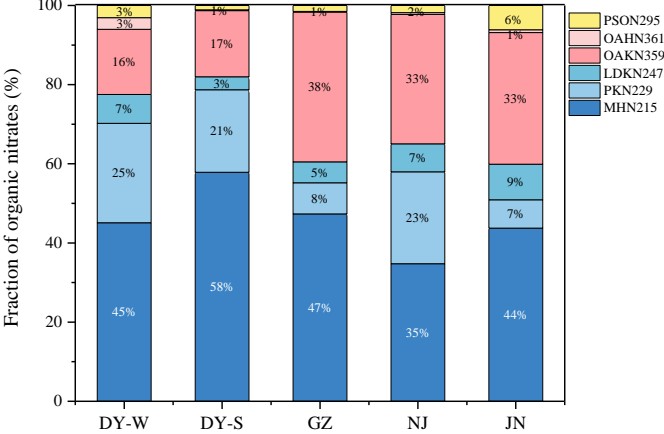

**Figure 2. Proportions of the six organic nitrates in the measured PONs at different sampling sites.**







**Figure 3. Daytime and nighttime concentrations of organic nitrates at different sampling sites.**





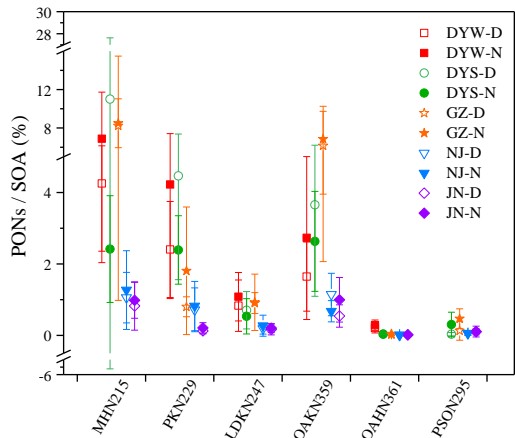


**Figure 4. Diurnal differences in PONs/SOA ratios for different organic nitrates and sampling sites.**

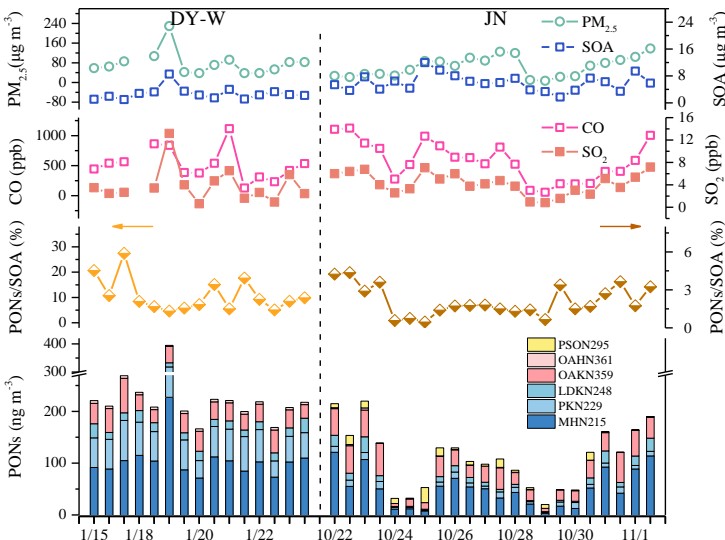

**Figure 5. Time series of PM$_{2.5}$, CO, SO$_2$, PONs/SOA, and PONs at DY site in winter and at JN site.**



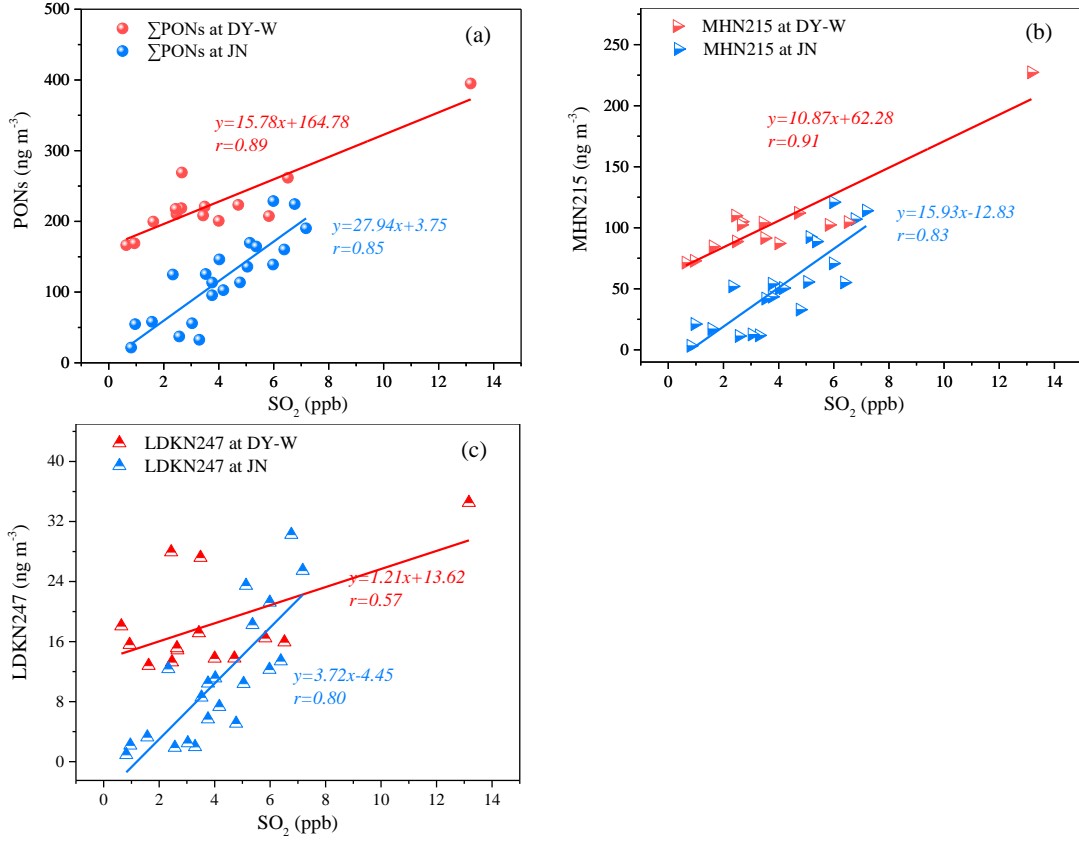

**Figure 6.** Correlation plots between SO$_2$ and (a) PONs, (b) MHN215, and (c) LDKN247 at DY site in winter and at JN sites.

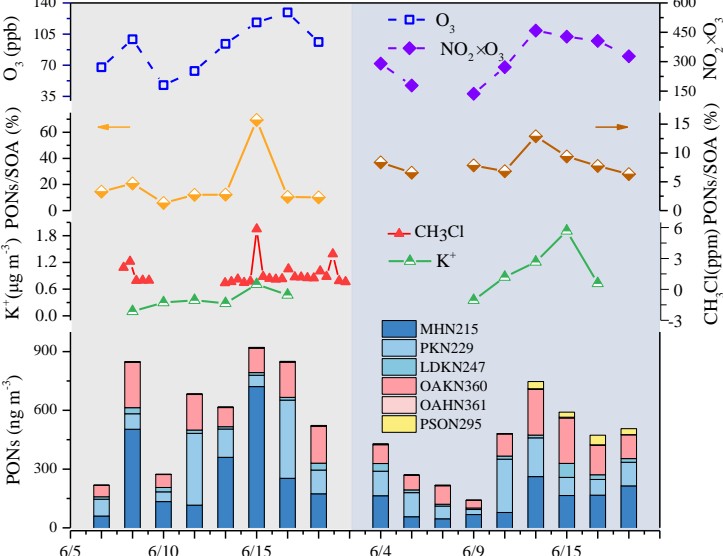

**Figure 7.** Time series of O$_3$, NO$_2$*O$_3$, PONs/SOA, PONs, CHCl$_3$, and K$^+$ at DY site in summer. The daytime and nighttime data are marked by gray and blue shading, respectively.



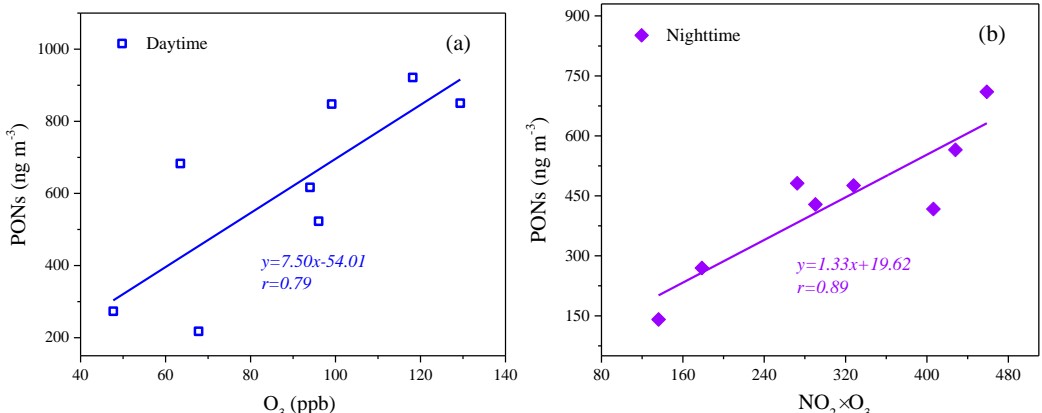


**Figure 8. Correlation plot of PONs with O$_3$ at daytime (a) and NO$_2$*O$_3$ at nighttime (b) at DY site in summer.**

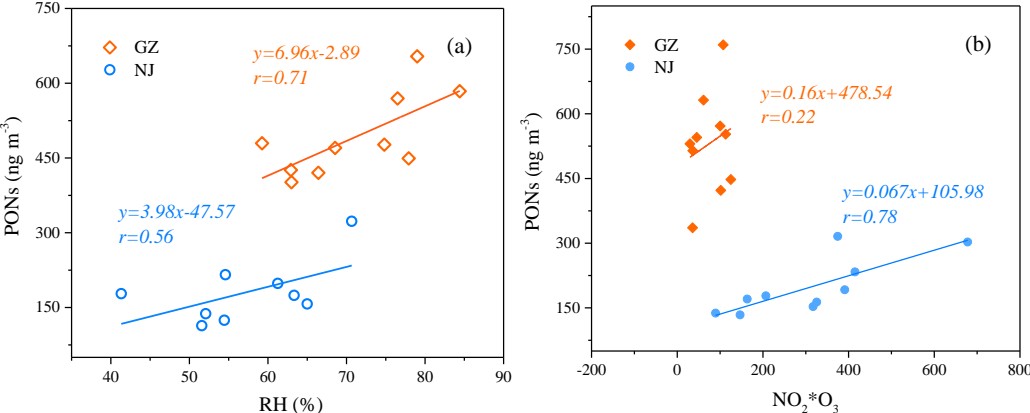

**Figure 9. Correlation plot of organic nitrates with (a) relative humidity at daytime, and with (b) the concentration product of O$_3$ and NO$_2$ at nighttime at GZ and NJ sites.**