# Peer review of "Particulate organic nitrates in eastern China: variation characteristics and effects of anthropogenic activities"

_Atmospheric Chemistry and Physics, 2019_

## Referee Comment (RC1) · Anonymous Referee #2 · 5 Nov 2019

Review of Zhang et al. entitled "Particulate organic nitrates in eastern china: variation characteristics and effects of anthropogenic activities"

This study presents the measurements of six different organic nitrates in the particle phase in places in China that are representative of both urban and rural environments. The levels and diurnal patterns of these nitrates are reported and their correlations with other factors such as meteorological conditions and anthropogenic tracers are discussed. Overall, the paper is clearly written and reports the measurements of a very important SOA constituent in China. However, one is left with an impression that the manuscript is simply reporting values and numbers other than a thorough analysis of the sources and transformations of these organic nitrates in the atmosphere. Prior to publication the authors should endeavor to provide a more mechanistic and quantitative discussion about various physical and chemical conditions/factors associated with the observed patterns of organic nitrates at different sampling sites.

General:

The manuscript has been largely improved in terms of the identification of organic nitrates using tandem mass spectrometry techniques. The observations of neutral losses of NO2 and HNO3 upon fragmentation of the parent ions certainly provide strong evidence for the presence of the -ONO2 functional group in the analytes of interest. The quantification of organic nitrates, however, is a big limiting factor in the accuracy and reliability of the data presented here. While I understand that the authentic standards for most nitrates studied here are not available (not all of them though, some monoterpene derived nitrates can be synthesized), using surrogates as an alternative could certainly bring large uncertainties to the measurements, which need to be assessed carefully. Specifically, one important question to answer is that is the electrospray ionization efficiency of the surrogates (depends on which functional group is ionized) on the same order of magnitude as the compound of interest?

The authors have spent almost the entire text describing the measurements of individual organic nitrates at different sampling sites: their concentrations during the day and nighttime, mass fractions in the total organic matters, and their correlations with other trace species like SO2. However, a mechanistic and quantitative exploration of the temporal variations and spatial distributions of different nitrates is lacking. A couple of examples:

1. The entire section of 3.2 'Diurnal difference of PONs' centers on discussing the different concentrations and patterns of organic nitrates in the day vs. night at different sampling sites. However, the mechanisms leading to such a pattern are barely explored. Figure 3 shows that the nighttime concentrations of MHN215 and PKN229 are only half of their daytime concentrations,

in contrast to almost all the other cases where the daytime and nighttime nitrate concentrations are quite comparable. It is well known that organic nitrates can be produced from NO3-initiated dark chemistry. A simple correlation analysis performed here does not provide any quantitative insights into the role of nighttime chemistry in the observed diurnal pattern of nitrates. The authors are suggested to do some calculations based on the NOx and O3 measurements to assess the intensities of nighttime chemistry (including both ozonolysis and NO3 oxidation) at different observational sites. Such a calculation is the first step to evaluate the contribution of nighttime chemistry to the production of organic nitrates.

2. The authors observed a positive correlation between organic nitrates and $SO_2$ and attributed such a correlation to that reactions of SO2 with stabilized Criegee intermediates produced from ozonolysis of monoterpenes could potentially promote the formation of organic nitrates. Such a conclusion is rather hasty and needs further observational evidence as support. The authors need to first consider: Are there any positive correlations between organic nitrates and other anthropogenic tracers such as NOx and CO? Are the temporal variations of organic nitrates simply a result of a sequence of pollution episodes caused by for example, high emissions or stagnant meteorological conditions? Furthermore, it is very important to keep in mind that the products from SCI+SO2 reactions are not necessarily low in vapor pressure and thus constituting potential precursors of SOA, not to mention that the stable products from SCI+SO2 reactions do not contain any -ONO2 functional groups and further oxidation steps are required to produce organic nitrates. Such a hypothetical reaction scheme is inconsistent with the chemical structures proposed for each nitrate listed in Table 1, which suggest that most organic nitrates identified, at least those monoterpene derived nitrates, are first generation products from OH/NO3 initiated oxidation of monoterpenes.

Specific:

Page 1, Line 25-30: Please carefully evaluate whether or not the SCI+SO2 reactions are associated with the observed nitrate production in the particle phase.

Page 2, Line 42: Clarify here oleic acid is mostly of anthropogenic origin.

Page 2, Line 49: Compounds with low vapor pressure are more likely condensing to particle phase and thus recognized as potential SOA precursor.

Page 2, Line 51: The references cited here are for the organic sulfate formation under acidic conditions, not nitrates.

Page 4, Line 139: Methanol is considered a very polar solvent, similar to water. Have the authors found any evidence of nitrate hydrolysis in pure methanol solvent? Have the authors tried to use

some nonpolar solvents such as acetonitrile? Any difference in terms of the quantification of organic nitrates?

Page 5, Line 160-165: Please explain why some nitrates have neutral losses of NO2 while others have HNO3 losses upon fragmentation?

Page 7, Line 280-285: Unless the measured nitrates are representative of unique tracers of coal burning aerosols, or other identified tracers of coal burning exhibited strong correlations with the nitrates, it is otherwise an invalid conclusion that the increase of PONs is due to coal burning activities.

Page 9, Line 330: The association of organic nitrate with the inorganic components in the particle phase does not necessarily establish any causal relationships. If it is the salting in effect as proposed by the authors, please provide some quantitative evidence.

Page 9, Line 360-365: Again, a positive correlation of RHs with organic nitrates in the particle phase does not necessarily establish any causal relationship. High RHs may just simply promote the photochemistry by generating more OH radicals, or high RHs is likely associated with a pollution episode. Keep in mind that the deliquescence RHs for various inorganic components in the particle phase are essential in determining the aerosol water content and its role in dissolving more water-soluble compounds (caution that some organic nitrates investigated in this study are not considered highly water soluble).

Page 18, Figure 2: In addition to the proportion of each organic nitrate species, please also show the total OM mass at different sampling sites.

Page 21, Figure 7: Please change the color code for PKN229 and LDKN247. It is hard to differentiate these two species based on the current blue hues.

---

## Referee Comment (RC2) · Anonymous Referee #1 · 6 Nov 2019

**Particulate organic nitrates in eastern China: variation characteristics and effects of anthropogenic activities**

**General comments**

This work conducted comprehensive analyses for organic nitrates in ambient $PM_{2.5}$ samples. The authors take advantage of the availability of $PM_{2.5}$ samples collected in four different sites in China and aim to identify major influential factors on organic nitrates formation in the different atmospheric environments through examining correlation relationship with organic nitrates and some co-emitted species.

However, this manuscript is just simply reporting the semi-quantified data of organic nitrates without any convincing and direct evidence to show the influence of anthropogenic pollutants to organic nitrates. This paper just provided non-robust discussion to support the previous reported results without any creative design and idea. Although the authors have the data from four different sites, the limited sample numbers cannot provide typical seasonal or diurnal information. They need a large dataset to prove their conclusions. Therefore, based on the significant weaknesses described above, publication is not recommended.

**Specific comments**

1. Line 114: Please provide sample numbers for each sampling site.

2. Line 176: The author mentioned that they employed surrogates to semi-quantify organic nitrates. The employment of surrogates that even don't contain the nitrooxy group would introduce large uncertainty for accurate quantification of organic nitrates because the different functional groups can introduce large differences for MS response. The authors even used different surrogates for different kind of organic nitrates but compared with the data together. The authors should find some methods to decrease the discrepancy or try to evaluate the uncertainty.

3. Line 184, the authors provided the recoveries for surrogate standards with only 60%. Based on my knowledge, the recoveries should in +- 20% to prove the effectiveness of an extraction method.

4. Line 188, please provide the comparison chromatogram between LC/orbitrap MS and LC/Iontrap MS.

5. Line 215, it mentioned: " increased emission of BVOCs in summer facilitated the secondary formation of PONSs." It's better to show some VOCs data, including such as isoprene, monoterpene and etc. If the authors didn't have detected data for their sites, maybe they can find some reported data near their sites. Still, the very limited number of samples in DY summer and winter may difficult to provide typical seasonal contrast (N~15). Because based on some work on monoterpene derived organosulfates, the formation of them may be more related to anthropogenic pollutions with a higher level in winter than in summer.

6. Line 243, the author detected the higher concentration of OAKN 359 (~100 ng/m$^3$), OAHN 361 based on surrogate standard and got the conclusion that significant influence of cooking and oil processing on the SOA in urban and rural areas in eastern China. First, the authors don't know the difference of MS response; therefore, it is possible that these compounds can have the large MS response but deficient concentration. Second, it's better to provide the MS/MS spectrum to prove MW 359 and MW 361 compounds share the same carbon skeleton with oleic acid.

7. In section 3.3.1, the author showed a strong correlation with $SO_2$ and LDKN 247 and got the conclusion that $SO_2$ promotes the formation of LDKN 247 through limonene CIs. First, please exclude the possibility of transportation that can lead to the strong correlation as well. Second, MW 247 organic nitrates can form from β-pinene (Clafin and Ziemann, JPCA, 2018) with two carbonyl groups, one hydroxyl group and one nitrooxy group. However, there's no research reported the β-pinene could react with $SO_2$ or $O_3$ to generate CIs. And the emission of β-pinene is larger than limonene.

8. Line 354, please provide the previous research to support that BB can lead to more VOCs emission.

9. Figure 5, "LDKN248" should be "LDKN247"

10. Figure 9(a), the number of blue points is 9. (b) the number of blue dots is 10.

11. Table 2, "(MW = 295, PSON 295" should be "(MW = 295, PSON 295)"

---

## Referee Comment (RC3) · Anonymous Referee #3 · 8 Nov 2019

Title: Particulate organic nitrates in eastern China: variation characteristics and effects of anthropogenic activities

Authors: Zhang et al.

The authors analyzed four organic nitrates in ambient $PM_{2.5}$ samples collected at four different sites in eastern China. In combination with other gaseous data, they aim to identify major controlling factors on organic nitrates formation under different atmospheric environments, mainly by examining the correlation relationship between organic nitrates and some gas species and meteorology parameters. Considering the high biogenic and anthropogenic emissions in eastern China, looking at the PNOs formation provides a good chance to explore the influence of human activities on biogenic SOA formation. However, due to limitations on the analyzing method and limited samples collected at each site and the different sampling periods, they are not able to provide a reliable seasonal/spatial/diurnal trend. While the discussion is heavily relying on correlation analysis, the factors controlling the observed PNOs are not sufficiently captured. Therefore, this paper can not be accepted in its current form.

General comments

1. While the topic is on the PNOs in eastern China, the authors only focused on six types of PNOs in the samples. As also mentioned in the discussion that the observed PNOs could be only a small fraction of the total PNOs. I wonder if the measured species here are the dominant PONs in the samples. Since the authors have conducted HRMS analysis, they should be able to clarify what are the major PNOs formulae in their samples and wether the focused species can represent the variation characteristics of the PNOs.

2. Several limitations should be considered regarding the analysis method. 1) Employment of surrogates that even don't contain the nitrooxy group would introduce large uncertainty for accurate quantification of PONs. 2) Compare data quantified with different surrogates introduce additional problems in the results and conclusion. As the focused point of this study is to explore the influence of anthropogenic emissions on the BSOA formation, comparison of different PNOs in concentration is not necessary. 3) Formic acid was added to the water only. As the eluent program develops, the formic acid in the mixture of the eluent will change. Formic acid was added to promote the ionization of the analytes. If the concentration of formic acid changes with time, we would expect ionization efficiency changes. This will introduce additional uncertainty in the semi-quantification. For examples, MHN215 and PSON 295 have several isomers which elute at different time. Due to the change in the formic acid concentration in the eluent, the ionization efficiency of different isomers can be different. 4) The authors reported a very low recovery efficiency of 60%. The uncertainty of the data should be evaluated carefully.

3. Though correlation analysis could provide some clues for explaining the observed season/diurnal/spatial trend, the actuall atmospheric process should also be considered.

The observed concentration of the PNOs in the particles is governed by the production, gas-particle portioning, the atmospheric loss process, and meteorological conditions . For example, the boundary layer height can play an important role in the observed diurnal trend of PONs. MHN215, PKN229, and LDK247 seem to be semivolatile organic compounds which will exist in both gas and particle phase. Thus the partitioning effect should be considered in the discussion. PONs can be partly photolyzed and have relatively short lifetime against hydrolysis. These may also be important in the explanation of the results.

Specific comments

1. Line 51-53. I don't get the logic to put this here. This is more related to the influence of $SO_2$ on BSOA formation rather than PONs production.

2. Line 65-70. Please arrange these sentences. Why put the description of the results in southern California here?

3. Line 134. Please clarify how these numbers are determined?

4. Line 161. Previous study by Clafine and Ziemann (2018) found that $NO_3$ radical oxidation of b-pinene produces di-keto nitrate.

5. It is not convincing by saying "The determination results from both mass spectrometers for the same samples showed good consistency". Please provide a chromatogram or overall mass spectra from the two instruments of the same sample.

6. Line 208-213. Lacking enough supporting for these discussions. First of all, the samples at four sites are collected at a different time of a year. Thus it is hard to obtain an accurate spatial trend. Secondly, as explained in the general comments, PONs are products from the interaction of biogenic and anthropogenic emissions. The emissions of VOCs and NOx, gas-particle partitioning, environmental conditions, and loss of PNOs should be considered in understanding the spatial and temporary trend.

7. Line224-232. Due to the limitation of the quantification, discussion on the composition/relative contribution of different PNOs is can hardly provide a concrete conclusion.

8. Line 234. I don't think in this paper the authors reported PNOs. Instead, the paper by He et al. 2014 and Lin et al. 2012 reported PSON in south China.

9. Line 242-245. These two arguments are not well supported. Since surrogate standard was used, the real concentration of acid-derived organic nitrates is hard to know. Though the oleic acid-derived organic nitrates were quantified with a considerable level, it is hard to estimate the contribution of cooking and oil processing on the SOA production from this single compound. Then you can not state like "significant influence". For the second argument on PSON295, there is no discussion on the production and loss at all. Thus, no conclusion can be drawn.

10. Line 250-256. The atmospheric bondary layer higher could also play an important role in the diurnal trend, especially for those that nighttime concentrations were higher.

11. Line 257-262. When discussing the seasonal trend, production due to emissions and reaction rate change can be important. However, the loss of PNOs due to photolysis and hydrolysis (Nah et al., 2016; Takeuchi and Ng, 2019) can be also different. These should also be considered.

12. Line 289-290. Back trajectory analysis for this day and other sampling days showing different transportation is very helpful to support this argument.

13. Line 297-300. While this study focused on the secondary produced PNOs in the atmosphere, it is hard to believe that the NO and $NO_2$ data has never been shown. For the discussion here, figures showing poor correlations between NOx and PONs are of equal importance as figures showing good correlation between PNOs and $SO_2$.

14. Line 312-314. If this is the contribution of the CI+$SO_2$ pathway, I will double that this can affect LDK247 production significantly. 1) in this calculation, lots of parameters are needed, such as $NO_3$ radical concentration, NO mixing ration, and $SO_2$ mixing ratio. The values being used in this calculation should be clarified, especially for NO which is never shown elsewhere. 2) to show the influence the $SO_2$, the author may do the calculation in an alternative way. That is to vary the $SO_2$ concentration in a reasonable range while fixing other parameters. Then the authors may see how can this pathway contribute to the LSK247 production.

15. Line 340-341. This is contradictory to the argument by the author that CI+SO2 have potential influence on PNOs formation. As elevated O3 concentration would promote CI production.

16. As I mentioned before, figures showing good correlation and poor correlation are of equal importance. This comparison provides us a more clear impression of the results.

17. Line 374-375. Please provide NO data to support this. Since NO is one of the key species in PNOS formation, I suggest the authors to put NO, NO2 data in table one.

Errors

1. Line 49, replace "higher" with "low"

2. Line 50. Change "a mechanism catalyzed by aerosol acidity" to "acid-catalyzed reactions".

3. Line 114. Change "contained" to "shown"

4. Line 167. Add "with" after "identified"

5. Line 304. Delete the space before PNOs

6. Figure 5, "LDKN248" should be "LDKN247"

7. Table 2, "(MW = 295, PSON 295" should be "(MW = 295, PSON 295)"

Reference

Claflin, M. S. and Ziemann, P. J.: Identification and Quantitation of Aerosol Products of the Reaction of β-Pinene with NO3 Radicals and Implications for Gas- and Particle-Phase Reaction Mechanisms, The Journal of Physical Chemistry A, 122, 3640-3652, 2018.

Lin, P., Yu, J. Z., Engling, G., and Kalberer, M.: Organosulfates in Humic-like Substance Fraction Isolated from Aerosols at Seven Locations in East Asia: A Study by Ultra-High-Resolution Mass Spectrometry, Environ. Sci. Technol., 46, 13118-13127, 2012.

Nah, T., Sanchez, J., Boyd, C. M., and Ng, N. L.: Photochemical Aging of α-pinene and β-pinene Secondary Organic Aerosol formed from Nitrate Radical Oxidation, Environ. Sci. Technol., 50, 222-231, 2016.

Takeuchi, M. and Ng, N. L.: Chemical Composition and Hydrolysis of Organic Nitrate Aerosol Formed from Hydroxyl and Nitrate Radical Oxidation of α-pinene and β-pinene, Atmos. Chem. Phys. Discuss., 2019, 1-39, 2019.

---

## Author Comment (AC1) · 28 Jan 2020

Review of Zhang et al. entitled "Particulate organic nitrates in eastern china: variation characteristics and effects of anthropogenic activities"

This study presents the measurements of six different organic nitrates in the particle phase in places in China that are representative of both urban and rural environments. The levels and diurnal patterns of these nitrates are reported and their correlations with other factors such as meteorological conditions and anthropogenic tracers are discussed. Overall, the paper is clearly written and reports the measurements of a very important SOA constituent in China. However, one is left with an impression that the manuscript is simply reporting values and numbers other than a thorough analysis of the sources and transformations of these organic nitrates in the atmosphere. Prior to publication the authors should endeavor to provide a more mechanistic and quantitative discussion about various physical and chemical conditions/factors associated with the observed patterns of organic nitrates at different sampling sites.

**Response:** we thank the reviewer for the critical comments and helpful suggestions. More discussion on physical and chemical mechanisms has been added in the revised manuscript to help to understand the observed variation patterns of monoterpene and oleic acid-derived particulate organic nitrates and the dominant influencing factors, including exploring the reasons for the diurnal difference, evaluating the impact of air mass transport, and supplementing additional evidences to the influencing factors. All of comments are responded in details as below.

General:

The manuscript has been largely improved in terms of the identification of organic nitrates using tandem mass spectrometry techniques. The observations of neutral losses of NO2 and HNO3 upon fragmentation of the parent ions certainly provide strong evidence for the presence of the -ONO$_2$ functional group in the analytes of interest. The quantification of organic nitrates, however, is a big limiting factor in the accuracy and reliability of the data presented here. While I understand that the authentic standards for most nitrates studied here are not available (not all of them though, some monoterpene derived nitrates can be synthesized), using surrogates as an alternative could certainly bring large uncertainties to the measurements, which need to be assessed carefully. Specifically, one important question to answer is that is the electrospray ionization efficiency of the surrogates (depends on which functional group is ionized) on the same order of magnitude as the compound of interest?

**Response:** thanks for the comments. It is rather difficult to determine the electrospray ionization efficiency of the surrogates, because the detachment cross section cannot be directly measured through the ESI/MS and the computing methods with regression models reported in some literatures are not

appropriate for multifunctional organic compounds. Therefore, we further test the response variations of the surrogate standards and ambient PONs in the mass spectrometer at different voltages to estimate the relative electrospray ionization efficiency. The ion source voltage is adjusted to 3000 V, 3500 V, 4000 V, and 4500 V. The results show that the variation in the mass spectrum response for surrogate standards and ambient PONs is relatively small when the voltage changes. Specifically, the average relative changes in the peak areas of the surrogate standard of (1R, 2R, 5R)–(+)-2-hydroxy-3-pinanone and monoterpene-derived organic nitrates are $11.2\% \pm 3.0\%$ and $11.4\% \pm 1.7\%$, respectively. The average relative changes in the peak areas of the surrogate standard of ricinoleic acid and oleic acid-derived nitrates are $9.3\% \pm 3.5\%$ and $12.4\% \pm 1.7\%$, respectively. The average relative changes in the peak areas of the surrogate standard of (-)-10-camphor sulfonic acid and pinene sulfate organic nitrate are $8.8\% \pm 7.8\%$ and $8.5\% \pm 7.0\%$, respectively.

In addition, the responses of n-pentane, amyl nitrate, isosorbide, and isosorbide 5-nitrate in the mass spectrometer are measured to estimate the differences in the response caused by the nitrate functional group. The difference in responses of n-pentane and amyl nitrate is $60.9\% \pm 7.2\%$, and the difference between isosorbide and isosorbide 5-nitrate is $66.6\% \pm 2.7\%$. Indeed, there is an inherent difference in responses of surrogate standards and analytes. However, the nitrate group does not cause a difference more than one order of magnitude. In this study, the structures and the retention time of the surrogate standards we selected are very close to those of analytes. Therefore, in our view the electrospray ionization efficiencies of surrogates and analytes are generally similar, and the concentrations of organic nitrates can be estimated reasonably and further used for exploring the variation characteristics.

Line 186-191, "*The difference in the mass spectrum response caused by the nitrate functional group was estimated with n-pentane, amyl nitrate, isosorbide, and isosorbide 5-nitrate. The difference between n-pentane and amyl nitrate was 60.9%±7.2%, and the difference between isosorbide and isosorbide 5-nitrate was 66.6%±2.7%. Indeed, there is an inherent difference in responses of surrogate standards and analytes, and the nitrate group does not cause a difference more than one order of magnitude. Therefore, the concentrations of organic nitrates can be estimated reasonably and further used for exploring the variation characteristics.*"

The authors have spent almost the entire text describing the measurements of individual organic nitrates at different sampling sites: their concentrations during the day and nighttime, mass fractions in the total organic matters, and their correlations with other trace species like SO2. However, a mechanistic and quantitative exploration of the temporal variations and spatial distributions of different nitrates is lacking. A couple of examples:

1. The entire section of 3.2 'Diurnal difference of PONs' centers on discussing the different concentrations and patterns of organic nitrates in the day vs. night at different sampling sites. However, the mechanisms leading to such a pattern are barely explored. Figure 3 shows that the nighttime concentrations of MHN215 and PKN229 are only half of their daytime concentrations, in contrast to

almost all the other cases wsim the daytime and nighttime nitrate concentrations are quite comparable. It is well known that organic nitrates can be produced from NO3-initiated dark chemistry. A simple correlation analysis performed here does not provide any quantitative insights into the role of nighttime chemistry in the observed diurnal pattern of nitrates. The authors are suggested to do some calculations based on the NOx and O3 measurements to assess the intensities of nighttime chemistry (including both ozonolysis and NO3 oxidation) at different observational sites. Such a calculation is the first step to evaluate the contribution of nighttime chemistry to the production of organic nitrates.

**Response:** the possible mechanisms leading to diurnal difference and the calculation results have been added in the revised manuscript.

Line 259–271, "*Particularly, at the DY site in summer, MHN215 and PKN229 exhibited obviously higher contributions at daytime than those at nighttime ($p < 0.01$). To verify whether the diurnal difference was aroused by the change in boundary layer height, the PONs/PM$_{2.5}$ ratios were also compared (see Fig. S3). The very similar diurnal trends between PON concentrations and PONs/PM$_{2.5}$ ratios suggest that the boundary layer change had little influence on the diurnal difference in the concentrations of PONs. Further cluster analysis (shown in Fig. S4) indicates that the air masses exhibited no significant difference between the daytime and nighttime sampling periods. It means that the diurnal difference was not mainly caused by air mass transport. Note that during daytime in summer at the DY site, biomass burning was intensive which can significantly increase the ozone concentration and enhance the atmospheric oxidation capacity (detailed discussion can be seen in Section 3.3.2). In addition, the emission rates of precursor monoterpenes were high at daytime due to the high temperature and the intense sunlight (Rinne et al., 2002). Therefore, the relative abundant oxidants and monoterpenes at daytime were ready to react to produce more monoterpenes-derived nitrates. For other kinds of organic nitrates, the concentrations during daytime and nighttime were comparable, that is consistent with the findings by Sobanski et al.(2017).*"

Line 281–287, "*To understand the role of nighttime chemistry playing in the abundances of organic nitrates, the nighttime production rate of organic nitrates was calculated via $P_{night} = \alpha(NO_3)k_{(NO2+O3)}[NO_2][O_3]$ (Sobanski et al., 2017), in which $\alpha(NO_3)$ is the yield of organic nitrates with $NO_3$, $k_{(NO2+O3)}$ is the rate coefficient for the reaction between $NO_2$ and $O_3$ and is determined by $1.4\times10^{-13}$ exp($-2470/T$) $cm^3$ molecule$^{-1}$ $s^{-1}$. The $\alpha(NO_3)$ reported in previous studies varies with precursors from 0.22 to 0.54 (Hallquist et al., 1999; Fry et al., 2009; Fry et al., 2011; Fry et al., 2014), and the mean value of 0.4 is adopted here. Based on the above calculations, the average nighttime prodution rates of PONs at the JN site and the DY site in summer were 381±400 ppt h$^{-1}$ and 362±157 ppt h$^{-1}$, respectively, which are fast enough to produce the observed PON concentrations.*"

[Figure]

**Figure S3.** The differences in PONs/PM$_{2.5}$ ratios during daytime and nighttime at different sampling

sites.

[Figure]

**Figure S4.** Air mass clusters classified by the 72-h backward trajectories at (a) daytime and (b) nighttime at the DY site in summer during the sampling period.

2. The authors observed a positive correlation between organic nitrates and SO2 and attributed such a correlation to that reactions of SO2 with stabilized Criegee intermediates produced from ozonolysis of monoterpenes could potentially promote the formation of organic nitrates. Such a conclusion is rather hasty and needs further observational evidence as support. The authors need to first consider: Are there any positive correlations between organic nitrates and other anthropogenic tracers such as NOx and CO? Are the temporal variations of organic nitrates simply a result of a sequence of pollution episodes caused by for example, high emissions or stagnant meteorological conditions? Furthermore, it is very important to keep in mind that the products from SCI+SO2 reactions are not necessarily low in vapor pressure and thus constituting potential precursors of SOA, not to mention that the stable products from SCI+SO2 reactions do not contain any -ONO2 functional groups and further oxidation steps are required to produce organic nitrates. Such a hypothetical reaction scheme is inconsistent with the chemical structures proposed for each nitrate listed in Table 1, which suggest that most organic nitrates identified, at least those monoterpene derived nitrates, are first generation products from OH/NO3 initiated oxidation of monoterpenes.

**Response:** at the DY site in winter, PONs was moderately correlated with CO (r = 0.61, $p < 0.01$). At the JN site the correlation between PONs and $NO_2$ or CO was negligible or weak (r = 0.17, p < 0.41 and r = 0.43, $p < 0.05$, respectively). On the basis of the observation data, the effect of air mass transport cannot be entirely ruled out. However, when compared with the strong correlation between $SO_2$ and

organic nitrates at the JN site and the DY site in winter (r=0.68 and 0.88, respectively; $p<0.01$), air mass transport is assumed to have no controlling effect on the variation of organic nitrates, especially at the JN site. Here, we further examine the correlation between the concentrations of $SO_2$ and PONs at the JN site after deducting the effect of transport. According to the linear fitting of the data of PONs and CO, the part of PONs which was associated with air mass transport is estimated by $[PONs]_{trans}=69.27\times[CO]+55.38$. Based on the observed concentrations of PONs and CO, the concentration of locally produced PONs is roughly calculated by subtracting the portion from air mass transport. The moderately strong correlations between estimated locally formed PONs and $SO_2$ (r=0.64; $p < 0.01$) confirm the large effect of coal combustion on the formation of PONs. In addition, the average concentrations of $PM_{2.5}$ at the JN site in autumn and at the DY site in winter were 69.5±42.5 ng/m$^3$ and 77.8±51.8 ng/m$^3$, respectively, with the ranges of 10.0–151.1 ng m$^{-3}$ and 37.8–229.4 ng m$^{-3}$, respectively. As indicated by the temporal variations of $PM_{2.5}$ in Figure 5, there were no continuous and severe pollution episodes during the sampling periods.

The reaction scheme of limonene with ozone, OH, and $NO_3$ to produce LDKN247 (i.e. HDCN247 in the revised manuscript) in the gas phase (shown in Fig. S7) is adopted from the Master Chemical Mechanism version 3.3.1 (MCM, http://mcm.leeds.ac.uk/MCM/). The gas-phase ozonolysis of limonene firstly generates CIs. The CI then reacts with $SO_2$ (or NO, $NO_2$, or CO) to form 3-isopropenyl-6-oxoheptanal, and finally produces LDKN247. The produced gas-phase LDKN247 is ready to partition into the particle phase. According to the predicted vapor pressure of 0.06 mmHg at 25 ℃ (predicted from the website of ChemSpider, available at http://www.chemspider.com/Chemical-Structure.2282909.html?rid=6fa367c1-85d1-4d19-83c2-e5fb2fb5466a), the product of 3-isopropenyl-6-oxoheptanal from SCI and $SO_2$ is semi-volatile and readily further reacts with $NO_3$ and OH at the rate constants of $2.6\times10^{-13}$ and $1.1\times10^{-10}$ at 298K, respectively. Therefore, the formation pathway of particulate organic nitrates from the reactions of CIs with $SO_2$ is feasible.

Line 334–342, "*At the DY site in winter, PONs was moderately correlated with CO (r = 0.61, p < 0.01, see Fig. 6 (c)), which indicates that the transport possibly had an effect on the concentration of PONs. However, the relationship between PONs and $NO_2$ or CO was negligible or weak (r = 0.17, p < 0.41 and r = 0.43, p < 0.05, respectively) at the JN site. It means that transport was not the controlling factor for the strong correlation between PONs and $SO_2$. With an assumption that the PONs were partly affected by transport and this portion was linked to the CO concentration, the PONs associated with air mass transport was roughly estimated by $[PONs]_{trans}=69.27\times[CO]+55.38$. The locally produced PONs was then approximated by subtracting the part from transport. As shown in Fig. 6 (d), the moderately strong correlation between locally formed PONs and $SO_2$ confirms the large effect of coal combustion on the formation of PONs during the measurement periods.*"

[Figure]

**Figure S7.** Formation mechanisms of HDCN247 from limonene adopted from the Master Chemical Mechanism (MCM).

[Figure]

[Figure]

**Figure 6**. Scatter plots and the correlations between PONs and (a) $SO_2$, (b) $NO_2$, and (c) CO and between (d) locally generated PONs and $SO_2$ at the DY site in winter and at the JN site.

Specific:

Page 1, Line 25-30: Please carefully evaluate whether or not the SCI+SO2 reactions are associated with the observed nitrate production in the particle phase.

**Response:** some additional discussion on the impact of $SO_2$ on PONs formation has been added in the revised manuscript (Section 3.3.1), and the related description has been clarified as follows.

Line 28–30, "Industrial and residential coal combustion played an important role in the increased concentrations of PONs through providing the related reactants and interfaces and promoting the formation rates of PONs."

Page 2, Line 42: Clarify here oleic acid is mostly of anthropogenic origin.

**Response:** oleic acid has been deleted from this sentence.

Line 41–45, "*The biogenic volatile organic compounds (BVOCs) isoprene, monoterpenes, and sesquiterpene have been identified as major precursors of PONs and the production yields are dramatically different with BVOC species (Fry et al., 2014; Boyd et al., 2015). Oleic acid, one of the most common fatty acid found in animals and plants, is recognized as a tracer for anthropogenic cooking aerosols and is also considered to be a precursor of organic nitrates (Docherty and Ziemann, 2006).*"

Page 2, Line 49: Compounds with low vapor pressure are more likely condensing to particle phase and thus recognized as potential SOA precursor.

**Response:** yes, it has been modified.

*Line 49–50, "Once formed, highly functionalized nitrates are expected to partition to the particle phase due to their low vapor pressure (Lim and Ziemann, 2005)."*

Page 2, Line 51: The references cited here are for the organic sulfate formation under acidic conditions, not nitrates.

**Response:** one of the compounds studied in Surratt et al. (2008) is nitrooxy organosulfate, i.e., pinene sulfate organic nitrate (PSON295) in our study, so we cited it here. Another reference of Han et al. (2016) indicated that particle acidity facilitated the formation of particulate organic nitrates. In the revised manuscript, the former reference has been removed and the latter reference is retained.

*Line 51–52, "Moreover, laboratory studies have shown that organic nitrates can be formed heterogeneously by acid-catalyzed reactions (Han et al., 2016)."*

Page 4, Line 139: Methanol is considered a very polar solvent, similar to water. Have the authors found any evidence of nitrate hydrolysis in pure methanol solvent? Have the authors tried to use some nonpolar solvents such as acetonitrile? Any difference in terms of the quantification of organic nitrates?

**Response:** the hydrolysis rate constant of organic nitrates will increase with solution acidity. During the hydrolysis of the α-pinene-derived nitrate, a proton is abstracted by water molecule and then final products are produced (Rindelaub et al., 2016). However, because of the electron donating effect of methyl, the H in the hydroxyl group of methanol is not as active as the H in water. Therefore, methanol is commonly used as the extraction solvent and eluent for organic compounds. In this study, no apparent hydrolysis was observed by using pure methanol solvent. Actually, we have used acetonitrile as potential solvent when we try to establish the analytical method; however, there is no significant difference in the extracting efficiency and the mass spectrum response between acetonitrile and methanol. In fact, the polarity of acetonitrile and methanol are very close. With consideration of the strong elution capacity for hydroxyl-containing substances and the low toxicity, methonal is eventually selected as the solvent to extract the organic nitrates in ambient particulate samples.

*Line 138–139, "Methanol and acetonitrile have similar polarities. Due to the low toxicity and good solubility for hydroxyl-containing compounds, methanol was eventually selected as the extraction solvent."*

Page 5, Line 160-165: Please explain why some nitrates have neutral losses of NO2 while others have HNO3 losses upon fragmentation?

**Response:** the different fragments are associated with the analytes' structures. The C-O bond is easier to break than the O-N bond, especially after adding the hydrogen to the hydroxyl group at β-carbon (see Fig. R1). As a result, it is common to see fragments of $NO_3$ and $HNO_3$ when the analyte contains nitrate groups. However, monoterpene-derived organic nitrates have denser electron clouds due to the ring structure, so the C-O bond is more stable than the O-N bond. For this reason, the O-N bond is ready to

break and produce the fragment of $NO_2$. The difference in fragments in this study is consistent with the mass spectrum analysis of oleic acid-derived organic nitrates and monoterpene-derived organic nitrates in previous studies by Docherty and Ziemann (2006) and Perraud et al. (2010), in which $HNO_3$ loss was found in the MS/MS spectra of oleic acid–derived organic nitrates and $NO_2$ losses was observed in the MS/MS spectra of the pinene-derived organic nitrate.

Figure R1. The molecular structure of the oleic acid hydroxyl nitrate

Page 7, Line 280-285: Unless the measured nitrates are representative of unique tracers of coal burning aerosols, or other identified tracers of coal burning exhibited strong correlations with the nitrates, it is otherwise an invalid conclusion that the increase of PONs is due to coal burning activities.

**Response:** $SO_2$ was used as a good tracer for coal combustion plumes. With examination of the backward trajectories (see Fig. S5) and the locations of coal combustion industries, it can be deduced that the elevated levels of $SO_2$ came from coal combustion emissions in urban and suburban Jinan. The strong correlation between $SO_2$ and PONs (see Fig. 6) indicates the elevated concentration of PONs was to some content associated with the coal combustion activities. At the DY site in winter, there are rare coal combustion industries near the sampling site. We suppose that the $SO_2$ was from the residential coal burning for heating demand. Here, several nitrated phenols which are abundant in coal combustion flue gas (Lu et al., 2019), are selected to further verify the influence of coal combustion. Some discussion has been added in the revised manuscript.

Line 320–322, "*In addition, several nitrated phenols, which are abundant in fresh coal combustion flue gas (Lu et al., 2019), also showed similar trends with $SO_2$ and PONs (Fig. S5), indicating the potential effect of coal combustion on the concentrations of PONs.*"

[Figure]

**Figure S5.** Time series of PONs, SO$_2$, 4-nitrophenol (4NP), and 5-nitrosalicylic acid (5NSA) at the DY site in winter.

Page 9, Line 330: The association of organic nitrate with the inorganic components in the particle phase does not necessarily establish any causal relationships. If it is the salting in effect as proposed by the authors, please provide some quantitative evidence.

**Response:** the salting in effect is cited from by Xu et al. (2015a) as we mentioned on Line 331 in the original manuscript. Our speculation is based on the similar properties of the target compounds and is put forward to provide a possible explanation. However, at present there is no feasible method to quantify the salting-in effect on PONs uptake. The related description has been removed in the revised manuscript.

Page 9, Line 360-365: Again, a positive correlation of RHs with organic nitrates in the particle phase does not necessarily establish any causal relationship. High RHs may just simply promote the photochemistry by generating more OH radicals, or high RHs is likely associated with a pollution episode. Keep in mind that the deliquescence RHs for various inorganic components in the particle phase are essential in determining the aerosol water content and its role in dissolving more water-soluble compounds (caution that some organic nitrates investigated in this study are not considered highly water soluble).

**Response:** the time series of RH and PONs have been shown in Figure S8. The possibility that the increased concentration of PONs was caused by OH enhancement or pollution events can be generally ruled out by the stable ozone and PM$_{2.5}$ concentrations. In our other filed measurements, a considerable concentration of MHN125 was found in cloud water samples, and thus the monoterpene-derived PONs is assumed to be soluble to a certain degree. The following discussion has been added in the revised manuscript.

Line 402–403, "*At the GZ site, the concentrations of gaseous pollutants and PM$_{2.5}$ were generally unchanged during the sampling periods, with no distinct pollution events.*"

Line 406–411, "*As the humidity increased, the fine particles were primarily in a liquid-like state during the daytime (Slade et al., 2019), and the secondary organic material was likely in aqueous phase when the humidity was higher than 60% (Song et al., 2015). In addition, when the ambient humidity was higher than the deliquescence points of inorganic components (40%-80%) (Martin, S. T, 2000), the emerging liquid water would affect the gas-aqueous partition and increase the heterogeneous uptake of precursors, oxidants, and products (Wang et al., 2017).*"

Line 412–414, "*It is possible that the epoxides undergo reasonably fast acid-catalyzed reactions at typical SOA acidities and produce a variety of organic nitrate species by nucleophilic reactions with nitrate (NO$_3^-$) (Mael et al., 2015), especially during daytime when the acidity of aerosols is strong (Behera et al., 2013).*"

Page 18, Figure 2: In addition to the proportion of each organic nitrate species, please also show the total OM mass at different sampling sites.

**Response:** the total OM mass concentrations have been added in Table 3.

Page 21, Figure 7: Please change the color code for PKN229 and LDKN247. It is hard to differentiate these two species based on the current blue hues.

**Response:** the color codes have been changed.

**References:**

Docherty, K. S., and Ziemann, P. J.: Reaction of oleic acid particles with NO3 radicals: Products, mechanism, and implications for radical-initiated organic aerosol oxidation, Journal of Physical Chemistry A, 110, 3567-3577, 10.1021/jp0582383, 2006.

Lu, C., Wang, X., Li, R., Gu, R., Zhang, Y., Li, W., Gao, R., Chen, B., Xue, L., and Wang, W.: Emissions of fine particulate nitrated phenols from residential coal combustion in China, Atmos. Environ., 203, 10-17, 10.1016/j.atmosenv.2019.01.047, 2019.

Perraud, V., Bruns, E. A., Ezell, M. J., Johnson, S. N., Greaves, J., and Finlayson-Pitts, B. J.: Identification of Organic Nitrates in the NO3 Radical Initiated Oxidation of α-Pinene by Atmospheric Pressure Chemical Ionization Mass Spectrometry, Environmental Science & Technology, 44, 5887-5893, 10.1021/es1005658, 2010.

Rindelaub, J. D., Borca, C. H., Hostetler, M. A., Slade, J. H., Lipton, M. A., Slipchenko, L. V., and Shepson, P. B.: The acid-catalyzed hydrolysis of an α-pinene-derived organic nitrate: kinetics, products, reaction mechanisms, and atmospheric impact, Atmospheric Chemistry and Physics, 16, 15425-15432, 10.5194/acp-16-15425-2016, 2016.

---

## Author Comment (AC2) · 28 Jan 2020

**General comments**

This work conducted comprehensive analyses for organic nitrates in ambient PM2.5 samples. The authors take advantage of the availability of PM2.5 samples collected in four different sites in China and aim to identify major influential factors on organic nitrates formation in the different atmospheric environments through examining correlation relationship with organic nitrates and some co-emitted species.

However, this manuscript is just simply reporting the semi-quantified data of organic nitrates without any convincing and direct evidence to show the influence of anthropogenic pollutants to organic nitrates. This paper just provided non-robust discussion to support the previous reported results without any creative design and idea. Although the authors have the data from four different sites, the limited sample numbers cannot provide typical seasonal or diurnal information. They need a large dataset to prove their conclusions. Therefore, based on the significant weaknesses described above, publication is not recommended.

**Response:** organic nitrates, produced via complex and ambiguous mechanisms, are important components of secondary organic aerosol. Since most authentic standards are not commercially available at present, it becomes a common quantitative method to capture the specific organic nitrate concentrations by using surrogate standards for research purposes. Currently, only a few articles presented the characteristics of organic nitrates in the real atmosphere at the molecular level, especially in polluted atmospheric environment in eastern China. In this work, we determined six kinds of monoterpene and oleic acid-derived organic nitrates and revealed their variation characteristics in the environment with high biogenic and anthropogenic emissions in eastern China. Further analyses with the assistant of backward trajectories and tracers for biomass burning and coal combustion confirm the effects of anthropogenic activities on organic nitrates. Coal combustion for industrial production and residential heating, via CIs+SO$_2$ and acid catalytic-reactions, possibly plays a greater role in the formation of organic nitrates than the previously assumed. In addition, 34 more samples have been supplemented in the revised manuscript to provide more reliable variation trends.

**Specific comments**

1. Line 114: Please provide sample numbers for each sampling site.

**Response:** the sample numbers have been added in Table 1.

2. Line 176: The author mentioned that they employed surrogates to semi-quantify organic nitrates. The employment of surrogates that even don't contain the nitrooxy group would introduce large uncertainty for accurate quantification of organic nitrates because the different functional groups can

introduce large differences for MS response. The authors even used different surrogates for different kind of organic nitrates but compared with the data together. The authors should find some methods to decrease the discrepancy or try to evaluate the uncertainty.

**Response:** the responses of n-pentane, amyl nitrate, isosorbide, and isosorbide 5-nitrate in the mass spectrometer have been measured. The results show that the difference in the response caused by the nitrate functional groups is smaller than one order of magnitude. Specifically, the difference in the response between n-pentane and amyl nitrate is 60.9% ±7.2% and the difference between isosorbide and isosorbide 5-nitrate is 66.6% ±2.7%. Indeed, there is an inherent difference in the responses of surrogate standards and analytes. However, the nitrate group does not cause a difference more than one order of magnitude. In this study, the structures and the retention time of the surrogate standards we selected are very close to those of analytes. Therefore, in our view the concentrations of organic nitrates can be estimated reasonably and further used for exploring the variation characteristics.

The contents of comparisons among PON concentrations quantified with different surrogates have been modified in the revised manuscript. The revised descriptions are as follows.

Line 202–205, "*Among these monoterpene-derived organic nitrates, MHN215 was the dominant species, with an average concentration of 51.9–212 ng m$^{-3}$, followed by PKN229 (10.0–113 ng m$^{-3}$), HDCN247 (9.5–21.6 ng m$^{-3}$) and PSON295 (3.1–9.6 ng m$^{-3}$). Among the oleic acid-derived organic nitrates, the average concentration of OAKN359 (34.1–146 ng m$^{-3}$) was much higher thanOAHN361 (0.7–4.1 ng m$^{-3}$).*"

Line 244–245, "*The large proportion of monoterpene-derived organic nitrates in PONs was also observed in southeastern United States (23–44% in mole) and in East Asia (Ayres et al., 2015; Lin et al., 2012)*"

The following description has been removed from the manuscript.

Line 225–226 in the original manuscript: "*MHN215 constituted the largest fraction of the identified PONs at all four sampling sites, accounting for 35–58%, followed by OAKN359 and PKN229.*"

In addition, we calculate the uncertainty of recovery and repeatability to measure the uncertainty in the detection. The results are as follows and have been added to the revised version.

Line 183–185, "*Based on the uncertainty in the recovery (±12.7%) and the repeatability (±4.1%), the overall uncertainty for the measurements was estimated to be 13.3%.*"

3. Line 184, the authors provided the recoveries for surrogate standards with only 60%. Based on my knowledge, the recoveries should in +- 20% to prove the effectiveness of an extraction method.

**Response:** the recoveries obtained in this study to some extent depend on the molecular type of the target compounds. In the previous study, the recoveries of some similar organic acid, such as pinic acid, pinonic acid, and camphor sulphonic acid, were about 74% (Kristensen and Glasius, 2011). In this study,

we have updated the recoveries for surrogate standards and evaluated the uncertainty based on a large number of recovery experiments. The following information has been updated and added in the revised manuscript.

Line 182–183, "*Recoveries of (1R, 2R, 5R)–(+)-2-hydroxy-3-pinanone, ricinoleic acid, and (-)-10-camphor sulfonic acid in six spiked samples were 71±9%, 64±4%, and 78±4%, respectively.*"

4.  Line 188, please provide the comparison chromatogram between LC/orbitrap MS and LC/Iontrap MS.

**Response:** the extracts of the collected PM$_{2.5}$ samples were analyzed partly by the LC/TSQ MS and partly by the LC/Orbitrap MS. The chromatograms have been added in Fig. S1 in the supporting information.

Line 191–195, "*In addition, the extracts of PM$_{2.5}$ samples were analyzed partly by the triple quadrupole tandem mass spectrometer and partly by the Orbitrap mass spectrometer (chromatograms can be seen in Fig. S1). The determination results from both mass spectrometers for the same samples showed good consistency, indicating the reliability of the quantifications of organic nitrates from the low-resolution triple quadrupole tandem mass spectrometer.*"

[Figure]

**Figure S1.** Chromatograms of ambient PM$_{2.5}$ samples for (a) ultra-performance liquid chromatography coupled with electrospray ionization and triple quadrupole tandem mass spectrometry and (b) ultra-high performance liquid chromatography equipped with electrospray ionization and orbitrap mass spectrometry.

5. Line 215, it mentioned: "increased emission of BVOCs in summer facilitated the secondary formation of PONSs." It's better to show some VOCs data, including such as isoprene, monoterpene and etc. If the authors didn't have detected data for their sites, maybe they can find some reported data near their sites. Still, the very limited number of samples in DY summer and winter may difficult to provide typical seasonal contrast (N~15). Because based on some work on monoterpene derived organosulfates, the formation of them may be more related to anthropogenic pollutions with a higher level in winter than in summer.

**Response:** At the DY site in winter and summer, we collected VOCs samples and subsequently detected the concentrations of isoprene. The data of isoprene has been shown in Fig. S2. Although the sampling periods of VOCs and PM$_{2.5}$ did not entirely overlap, it can still indicate the seasonal difference in BVOCs emission and the ambient concentration at the DY site, i.e., the concentration of isoprene in summer is 19 times higher than that in winter.

After additional data of PM$_{2.5}$ samples were supplemented, there were 26 and 21 samples at the DY site in summer and winter, respectively. Here we wanted to demonstrate that two factors likely affected the formation and concentrations of organic nitrates. One factor is "*the intense photochemical activities (e.g. 71.2 ppb of ozone)*" which was linked to biomass burning and was discussed in detail in section 3.3.2. The other factor is "*the increased emission of BVOCs*" which was partially caused by biomass burning and partially due to the abundant biological resources and high temperature in summer.

Line 225–226, "*the intense photochemical activities (e.g. 71.2 ppb of ozone) and increased emission of BVOCs in summer (see Fig. S2) facilitated the secondary formation of PONs.*"

[Figure]

**Figure S2.** Comparison of isoprene concentrations in winter and summer at the DY site.

6. Line 243, the author detected the higher concentration of OAKN 359 (~100 ng/m$^3$), OAHN 361 based on surrogate standard and got the conclusion that significant influence of cooking and oil processing on the SOA in urban and rural areas in eastern China. First, the authors don't know the difference of MS response; therefore, it is possible that these compounds can have the large MS response but deficient concentration. Second, it's better to provide the MS/MS spectrum to prove MW 359 and MW 361 compounds share the same carbon skeleton with oleic acid.

**Response:** the related sentence has been modified as follows.

Line 223–225, "*The identification and quantification of oleic acid-derived organic nitrates in this study confirm that the impact of cooking and oil processing on SOA in urban areas is more significant than that in rural areas in eastern China.*"

As shown in Figure R2, the MS/MS spectrum of MW 359 exhibits signal peaks at m/z 342 and m/z 268 and 264 from the molecular ion and the fragments which are formed by loss of $H_2O$ and $HNO_3+C_2H_5$, respectively. The MS/MS spectrum of MW 361 exhibits signal peaks at m/z 344, m/z 300, and m/z 256 which are formed by loss of $H_2O$, $NO_3$, and $NO_3+CO_2$, respectively. Similarly, as reported by the Human Metabolome Database (HMDB) (Wishart et al. 2017) (see Fig. R2), the MS/MS spectrum of ricinoleic acid exhibits peaks at m/z 279 and m/z 235 which are formed by loss of $H_2O$ and $CO_2$, respectively. In spite of the above MS/MS spectrum information, it is rare to use the MS/MS spectrum to confirm the carbon skeleton, because the MS/MS spectrum mainly provides the information of characteristic functional groups. In addition, it is not suitable to determine the carbon skeleton by nuclear magnetic resonance (NMR), because the sample solutions are mixed with many kinds of organic compounds. In this study, the structure of MW 359 and MW 361 are identified by the molecular weights and the characteristic fragments reported by Docherty and Ziemann (2006).

[Figure]

[Figure]

Figure R2. The MS/MS spectrum of (a) MW 359, (b) MW 361, and (c) ricinoleic acid (Wishart et al., 2017).

7. In section 3.3.1, the author showed a strong correlation with $SO_2$ and LDKN 247 and got the conclusion that $SO_2$ promotes the formation of LDKN 247 through limonene CIs. First, please exclude the possibility of transportation that can lead to the strong correlation as well. Second, MW 247 organic nitrates can form from β-pinene (Clafin and Ziemann, JPCA, 2018) with two carbonyl groups, one hydroxyl group and one nitrooxy group. However, there's no research reported the β-pinene could react with $SO_2$ or $O_3$ to generate CIs. And the emission of β-pinene is larger than limonene.

**Response:** we estimated the concentration of PONs after removal of transportation impact, the moderately strong correlation between the estimated locally formed PONs and $SO_2$ confirm the large impact of coal combustion on the formation of PONs. The estimation method and the added discussion are as follows.

Line 334–342, "*At the DY site in winter, PONs was moderately correlated with CO (r = 0.61, p < 0.01, see Fig. 6 (c)), which indicates that the transport possibly had an effect on the concentration of PONs. However, the relationship between PONs and $NO_2$ or CO was negligible or weak (r = 0.17, p < 0.41 and r = 0.43, p < 0.05, respectively) at the JN site. It means that transport was not the controlling factor*

*for the strong correlation between PONs and SO₂. With an assumption that the PONs were partly affected by transport and this portion was linked to the CO concentration, the PONs associated with air mass transport was roughly estimated by $[PONs]_{trans}=69.27\times[CO]+55.38$. The locally produced PONs was then approximated by subtracting this part from transport. As shown in Fig. 6 (d), the moderately strong correlation between locally formed PONs and SO₂ confirms the large effect of coal combustion on the formation of PONs during the measurement periods.*"

8. We thank the reviewer for the helpful information about another possible generation pathway of LDKN 247. As stated in Clafin and Ziemann (2018), they identified the formation of LDKN 247 from β-pinene which was initiated by $NO_3$ in an environmental chamber, with a mole fraction of 7.7%. Noticeably, the proportion of LDKN 247 is not high and it may be even lower in the real complex atmosphere. Another limitation for the formation of LDKN 247 from β-pinene is that only nocturnal process was studied and there is no evidence for the daytime generation. Therefore, it still needs further investigations to confirm the contribution to LDKN 247 in the ambient air. The hypothesis that we propose here for the reaction of $SO_2$ and pinene-derived CIs is based on the similar ring structure and carbon-carbon double bonds of pinene and limonene. Furthermore, many organic nitrates detected in our study are derived from both α- and β-pinene, and the emission of α-pinene is much higher than β-pinene (Cheng et al., 2018). In the revised manuscript, the reference of Clafin and Ziemann (2018) has been added in Table 2 and the name of LDKN 247 has been changed to hydroxydicarbonyl nitrate (HDCN 247) in order to avoid misleading.Line 354, please provide the previous research to support that BB can lead to more VOCs emission.

**Response:** as shown in Figure R3, higher concentrations of ambient isoprene were observed during the periods of biomass burning than usual, indicating the intensive emissions of BVOCs from biomass burning. Two related references have been added in the revised manuscript.

Line 397–398, "*biomass burning also led to increased emissions of BVOCs (Andreae and Merlet, 2001; Xu et al., 2018)*"

[Figure]

Figure R3. The concentration of isoprene at DY site in summer

9. Figure 5, "LDKN248" should be "LDKN247"

**Response:** thanks for the correction. LDKN 247 has been changed into the new name of "HDCN 247" in the revised manuscript.

10. Figure 9(a), the number of blue points is 9. (b) the number of blue dots is 10.

**Response:** Figure 9(a) shows the data points during daytime, while 9(b) shows data points at nighttime. The reason why the numbers of data points are different in the two figures is that the numbers of the daytime and nighttime samples are not equal.

11. Table 2, "(MW = 295, PSON 295" should be "(MW = 295, PSON 295)"

**Response:** the omission of the bracket has been added.

**References:**

Cheng, X., Li, H., Zhang, Y., Li, Y., Zhang, W., Wang, X., Bi, F., Zhang, H., Gao, J., Chai, F., Lun, X., Chen, Y., Gao, J., and Lv, J.: Atmospheric isoprene and monoterpenes in a typical urban area of Beijing: Pollution characterization, chemical reactivity and source identification, Journal of Environmental Sciences, 71, 150-167, https://doi.org/10.1016/j.jes.2017.12.017, 2018.

Docherty, K. S., and Ziemann, P. J.: Reaction of oleic acid particles with NO3 radicals: Products, mechanism, and implications for radical-initiated organic aerosol oxidation, Journal of Physical Chemistry A, 110, 3567-3577, 10.1021/jp0582383, 2006.

Kristensen, K., and Glasius, M.: Organosulfates and oxidation products from biogenic hydrocarbons in fine aerosols from a forest in North West Europe during spring, Atmospheric Environment, 45, 4546-4556, https://doi.org/10.1016/j.atmosenv.2011.05.063, 2011.

Wishart, D. S., Feunang, Y. D., Marcu, A., Guo, A. C., Liang, K., Vázquez-Fresno, R., Sajed, T., Johnson, D., Li, C., Karu, N., Sayeeda, Z., Lo, E., Assempour, N., Berjanskii, M., Singhal, S., Arndt, D., Liang, Y., Badran, H., Grant, J., Serra-Cayuela, A., Liu, Y., Mandal, R., Neveu, V., Pon, A., Knox, C., Wilson, M., Manach, C., and Scalbert, A.: HMDB 4.0: the human metabolome database for 2018, Nucleic Acids Research, 46, D608-D617, 10.1093/nar/gkx1089, 2017.

---

## Author Comment (AC3) · 28 Jan 2020

**Responses to review comments-3 (RC3, by the Anonymous Referee #3)**

Title: Particulate organic nitrates in eastern China: variation characteristics and effects of anthropogenic activities

Authors: Zhang et al.

The authors analyzed four organic nitrates in ambient PM2.5 samples collected at four different sites in eastern China. In combination with other gaseous data, they aim to identify major controlling factors on organic nitrates formation under different atmospheric environments, mainly by examining the correlation relationship between organic nitrates and some gas species and meteorology parameters. Considering the high biogenic and anthropogenic emissions in eastern China, looking at the PNOs formation provides a good chance to explore the influence of human activities on biogenic SOA formation. However, due to limitations on the analyzing method and limited samples collected at each site and the different sampling periods, they are not able to provide a reliable seasonal/spatial/diurnal trend. While the discussion is heavily relying on correlation analysis, the factors controlling the observed PNOs are not sufficiently captured. Therefore, this paper can not be accepted in its current form.

**Response:** Thanks for the critical comments. It is very difficult to measure the individual organic nitrates and to get accurate measurement results due to the lack of authentic standards. Although the use of surrogate standards will add uncertainty to the absolute concentration, these data can still provide important information on the variation trends. In the revised manuscript, more information has been added to increase the reliability of the analyzing method and 34 additional samples have been supplemented to provide more reliable variation trends. The influences from boundary layer change, air mass transport, and gas-particle partition have been discussed to obtain credible conclusions.

General comments

1. While the topic is on the PNOs in eastern China, the authors only focused on six types of PNOs in the samples. As also mentioned in the discussion that the observed PNOs could be only a small fraction of the total PNOs. I wonder if the measured species here are the dominant PONs in the samples. Since the authors have conducted HRMS analysis, they should be able to clarify what are the major PNOs formulae in their samples and whether the focused species can represent the variation characteristics of the PNOs.

**Response:** thanks for the comment. With the assistant of high-resolution mass spectrometry, some other potential organic nitrates are noticed, and the possible molecular formulas include $C_5H_9NO_4$, $C_6H_{13}NO_4$, $C_7H_{15}NO_4$, $C_8H_9NO_4$, $C_9H_{11}NO_4$, $C_9H_{19}NO_4$, and $C_{11}H_{23}NO_4$. The raw signals of these potential organic nitrates are comparable to or lower than those of the targeted organic nitrates in this study. In addition, a

newly published paper by Shi et al. (2020) has identified dozens of kinds of organic nitrates by using ToF-MS. However, most of the structures have not been confirmed and the approximate concentrations have not been quantified. In our previous unpublished study, we have tried to determine the total organic nitrate by using flourier transform infrared spectroscopy. According to the preliminary results, these six organic nitrates accounted for 15% to 75% of the total organic nitrates in different seasons. With consideration of that the fractions of the six species in total organic nitrates are not very high, the title of this manuscript has changed to "*Monoterpene and oleic acid-derived particulate organic nitrates in eastern China: variation characteristics and effects of anthropogenic activities*". The relevant sentences in the abstract and the main text have been also changed correspondingly.

**2.** Several limitations should be considered regarding the analysis method. 1) Employment of surrogates that even don't contain the nitrooxy group would introduce large uncertainty for accurate quantification of PONs. 2) Compare data quantified with different surrogates introduce additional problems in the results and conclusion. As the focused point of this study is to explore the influence of anthropogenic emissions on the BSOA formation, comparison of different PNOs in concentration is not necessary. 3) Formic acid was added to the water only. As the eluent program develops, the formic acid in the mixture of the eluent will change. Formic acid was added to promote the ionization of the analytes. If the concentration of formic acid changes with time, we would expect ionization efficiency changes. This will introduce additional uncertainty in the semi-quantification. For examples, MHN215 and PSON 295 have several isomers which elute at different time. Due to the change in the formic acid concentration in the eluent, the ionization efficiency of different isomers can be different. 4) The authors reported a very low recovery efficiency of 60%. The uncertainty of the data should be evaluated carefully.

**Response:** 1) Thanks for the comment. The responses of n-pentane and amyl nitrate, isosorbide and isosorbide 5-nitrate in the mass spectrometer have been further determined to estimate the difference in the response caused by the nitrate functional groups. The determination results show that the difference in the response between n-pentane and amyl nitrate is 60.9±7.17% and that between isosorbide and isosorbide 5-nitrate is 66.6±2.69%. That is, there is indeed an inherent difference between the responses of the nitrate-free surrogate standard and the organic nitrate analyte in the mass spectrometer. However, the discrepancy is below one order of magnitude. In this study, the surrogate standards we selected are quite similar to the organic nitrate analytes in structure and property (indicated by the close retention time), and the nitrate group does not cause an order of magnitude difference. Therefore, the surrogate standards can semi-quantitate the analyte and provide the information on the variation trends.

Line 186–191, "*The difference in the mass spectrum response caused by the nitrate functional group was estimated with n-pentane, amyl nitrate, isosorbide, and isosorbide 5-nitrate. The difference between n-pentane and amyl nitrate was 60.9%±7.2%, and the difference between isosorbide and isosorbide 5-nitrate was 66.6%±2.7%. Indeed, there is an inherent difference in responses of surrogate standards and analytes, and the nitrate group does not cause a difference more than one order of magnitude.*

*Therefore, the concentrations of organic nitrates can be estimated reasonably and further used for exploring the variation characteristics.*"

2) The comparisons of PON concentrations quantified with different surrogates have been modified in the modified or removed in the revised manuscript. The revised descriptions are as follows.

Line 202-205: "*Among these monoterpene-derived organic nitrates, MHN215 was the dominant species, with an average concentration of 51.9–212 ng m$^{-3}$, followed by PKN229 (10.0–113 ng m$^{-3}$), HDCN247 (9.5–21.6 ng m$^{-3}$) and PSON295 (3.1–9.6 ng m$^{-3}$). Among the oleic acid-derived organic nitrates, the average concentration of OAKN359 (34.1–146 ng m$^{-3}$) was much higher than OAHN361 (0.7–4.1 ng m$^{-3}$).*"

Line 244-245: "*The large proportion of monoterpene-derived organic nitrates in PONs was also observed in southeastern United States (23–44% in mole) and in East Asia (Ayres et al., 2015; Lin et al., 2012)*"

The following description has been removed from the manuscript.

Line 225–226 in the original manuscript: "*MHN215 constituted the largest fraction of the identified PONs at all four sampling sites, accounting for 35–58%, followed by OAKN359 and PKN229.*"

3) Thank the reviewer's useful comments. To make clear the difference in the responses of different isomers caused by the change in eluent compositions, we have tested five samples to compare the difference in the response under different eluent conditions. Specifically, formic acid is added in both mobile phases of methanol and water with a concentration of 0.1‰, which is close to the concentration of formic acid when MHN215 is eluted in the positive ion mode. In addition, 36‰ formic acid is also added to the mobile phases, which is similar to the situation when PSON295 is eluted in the negative ion mode.

For the responses of the two isomers of MHN215, the relative mean deviation under the above two eluent conditions is 4.1%–2.9%. Due to the close retention time of the two isomers and the small gradient change when it was eluted, there is no significant difference in the response in the mass spectrometer and thus no obvious uncertainty in the measured concentrations of MHN215. As to PSON 295, the average relative deviation of the response of the first eluted isomer under two test conditions is 29.7±6.7%. The relative mean deviation of the second isomers is 13.4±6.7%. The slightly high difference for the two isomers of PSON 295 may be due to the relative large discrepancy in the elution time. Despite all this, it will not significantly change the relevant results and conclusion in this study, because the concentrations of PSON 295 are very low.

4) The recoveries obtained in this study to some extent depend on the molecular type of the target compounds. In the previous study, the recoveries of some similar compounds, such as pinic acid, pinonic acid, and camphor sulphonic acid, were about 74% (Kristensen and Glasius, 2011). In this study, we

have updated the recoveries for surrogate standards and evaluated the uncertainty based on a large number of recovery experiments. The following information has been updated and added in the revised manuscript.

Line 182–185, "*Recoveries of (1R, 2R, 5R)–(+)-2-hydroxy-3-pinanone, ricinoleic acid, and (-)-10-camphor sulfonic acid in six spiked samples were 71±9%, 64±4%, and 78±4%, respectively. Based on the uncertainty in the recovery (±12.7%) and the repeatability (±4.1%), the overall uncertainty for the measurements is estimated to be 13.3%.*"

3. Though correlation analysis could provide some clues for explaining the observed season/diurnal/spatial trend, the actual atmospheric process should also be considered. The observed concentration of the PNOs in the particles is governed by the production, gas-particle portioning, the atmospheric loss process, and meteorological conditions. For example, the boundary layer height can play an important role in the observed diurnal trend of PONs. MHN215, PKN229, and LDK247 seem to be semivolatile organic compounds which will exist in both gas and particle phase. Thus the partitioning effect should be considered in the discussion. PONs can be partly photolyzed and have relatively short lifetime against hydrolysis. These may also be important in the explanation of the results.

**Response:** Thanks for the helpful comments. Additional discussion on the boundary layer height change, gas-particle partitioning effect, and the losses processes has been added in the revised manuscript to explain the variation patterns of the observed particulate organic nitrates.

Line 260–264, "*To verify whether the diurnal difference was aroused by the change in boundary layer height, the PONs/PM$_{2.5}$ ratios were also compared (see Fig. S3). The very similar diurnal trends between PON concentrations and PONs/PM$_{2.5}$ ratios suggest that the boundary layer change had little influence on the diurnal difference in the concentrations of PONs. Further cluster analysis (shown in Fig. S4) indicates that the air masses exhibited no significant difference between the daytime and nighttime sampling periods. It means that the diurnal difference was not mainly caused by air mass transport.*"

Line 274–278, "*At DY site in winter, the emission of BVOCs was not high in cold season and the high production of PONs was attributed that the organic nitrates readily partition into the particle phase from the gas phase at low temperature. As reported in the previous study, the production of SOA was estimated to increase by more than 20% when the temperature dropped by 10 ℃ (Sheehan et al., 2001).*"

Line 288–298, "*On the basis of the observation data, it was difficult to see any apparent effects of photolysis and hydrolysis on the concentrations and variations of PONs from temperature and humidity. Takeuchi and Ng (2019) comprehensively evaluated the hydrolysis processes of particulate organic nitrates which were generated from the oxidation of α-pinene and β-pinene and the results showed that the hydrolysis lifetime was fast, within 30 min. However, in this study, the concentrations of PONs and*"

*their ratios to PM$_{2.5}$ in the moist conditions in summer were even higher than those in another season. With consideration of the moderate humidity in northern China, hydrolysis was not considered as the dominant factor that caused the lower concentrations of PONs at nighttime than daytime at the DY site in summer. In addition, the concentrations of PONs and their ratios to PM$_{2.5}$ in summer with intensive solar radiation were not lower than those in other seasons. Furthermore, the organic nitrates produced from different precursors have different photochemical aging behaviors and the photolysis rates of the organic nitrates are relatively small when compared to their production rates (Nah et al., 2016; Luke et al., 1989), so photolysis was not the major factor that governed the diurnal differences of PONs in this study.*"

Specific comments

1. Line 51-53. I don't get the logic to put this here. This is more related to the influence of SO2 on BSOA formation rather than PONs production.

**Response:** one of the compounds studied in Surratt et al. (2008) is nitrooxy organosulfate, i.e., pinene sulfate organic nitrate (PSON295) in our study, so we cited it here in the initial manuscript. Another reference of Han et al. (2016) indicates that particle acidity facilitated the formation of particulate organic nitrates. In the revised manuscript, the former reference has been removed to avoid potential misunderstanding, and the latter reference is retained.

Line 51–52, "*Moreover, laboratory studies have shown that organic nitrates can be formed heterogeneously by acid-catalyzed reactions (Han et al., 2016).*"

2. Line 65-70. Please arrange these sentences. Why put the description of the results in southern California here?

**Response:** here is an example about the impacts of specific human activities on the ambient concentrations of organic nitrates after giving the general description. The related sentences have been revised.

Line 64–66, "*Day et al. (2010) found that mixed urban fossil fuel combustion was primarily responsible for the high concentrations of PONs in southern California. On the basis of latest observations of PONs in China, large disparity exists and the concentrations of PONs range from below detection limit to several micrograms per cubic meter*".

3. Line 134. Please clarify how these numbers are determined?

**Response:** the ratios of SOA to SOC were assumed to the same with the ratios of organic matters to OC reported in previous studies in the references, which were calculated by the measured OC concentrations and the measured or calculated concentrations of organic matters. The above information has been added in the revised manuscript,

Line 130–134, "*Furthermore, according to the reported ratios of organic matters (OM) to OC based on*

*measurements and calculations in previous studies (Turpin and Lim, 2001; Liu et al., 2017), the OM concentrations were estimated from the OC concentrations and the SOA concentrations were estimated from the SOC concentrations by multiplying by a factor to give an estimate, i.e. 2.1, 2.2, 1.8, 1.6 and 1.6 for DY site in winter, DY site in summer, GZ, NJ, and JN sites, respectively.*"

4. Line 161. Previous study by Clafine and Ziemann (2018) found that NO3 radical oxidation of β-pinene produces di-keto nitrate.

**Response:** thanks for the reminder. The reference of Clafin and Ziemann (2018) has been added in Table 2 and the name of LDKN 247 has been changed into hydroxydicarbonyl nitrate (HDCN 247) to avoid misleading.

5. It is not convincing by saying "The determination results from both mass spectrometers for the same samples showed good consistency". Please provide a chromatogram or overall mass spectra from the two instruments of the same sample.

**Response:** the chromatograms from two instruments have been shown in Fig. S1 in the supporting information.

Line 191–195, "*In addition, the extracts of PM$_{2.5}$ samples were analyzed partly by the triple quadrupole tandem mass spectrometer and partly by the Orbitrap mass spectrometer (chromatograms can be seen in Fig. S1). The determination results from both mass spectrometers for the same samples showed good consistency, indicating the reliability of the quantifications of organic nitrates from the low-resolution triple quadrupole tandem mass spectrometer.*"

[Figure]

[Figure]

**Figure S1.** Chromatograms of ambient PM$_{2.5}$ samples for (a) ultra-performance liquid chromatography coupled with electrospray ionization and triple quadrupole tandem mass spectrometry and (b) ultra-high performance liquid chromatography equipped with electrospray ionization and orbitrap mass spectrometry.

6. Line 208-213. Lacking enough supporting for these discussions. First of all, the samples at four sites are collected at a different time of a year. Thus it is hard to obtain an accurate spatial trend. Secondly, as explained in the general comments, PONs are products from the interaction of biogenic and anthropogenic emissions. The emissions of VOCs and NOx, gas-particle partitioning, environmental conditions, and loss of PNOs should be considered in understanding the spatial and temporary trend.

**Response:** thanks for the good comments. It has been deleted that "*A comparison of different sampling sites shows that elevated PONs concentrations usually appeared at the regions with high vegetation cover.*" Instead, the manuscript been revised to try to only explain the reasons why high concentrations of PONs appeared at the GZ site and the DY site in summer. The emissions of precursors, partitioning, hydrolysis and photolysis have been discussed in the revised manuscript as follows.

Line 215–220, "*With examination of related pollutants and parameters, an observable increase in the concentration of PONs was noticed with the rising OM concentration at the GZ site and at the DY site in summer. However, the OM concentrations at the two sites were not very high and they did not increase proportionally with the concentration of PONs. With consideration of the high concentrations of BVOCs at the DY site in summer (see Fig. S2) and the intense photochemical activities (71.2 ppb of ozone on average), we attribute the enhanced formation of monoterpene and oleic acid-derived PONs (indicated by elevated ratios of ΣPONs/SOA) at the DY site primarily to the increase of precursors and oxidants.*"

Line 221–223, "*The high levels of monoterpene and oleic acid-derived PONs and the higher ratios to PM$_{2.5}$ and SOA at GZ site than was primarily ascribed to the intensive emissions of BVOCs in summer from massive broad-leaf trees in the subtropical region (Zheng et al., 2000).*"

Line 226–229, "*In this study, we did not see obvious evidence that photolysis or hydrolysis affected the concentration of organic nitrates. Nevertheless, the variability of the concentrations of PONs indicates that ambient temperature had non-observable effect and humidity had a positive effect on the*

*concentrations of PONs at the GZ site, which is not in favor of the hydrolysis loss.*"

7. Line224-232. Due to the limitation of the quantification, discussion on the composition/relative contribution of different PNOs is can hardly provide a concrete conclusion.

**Response:** the comparisons of the compositions and relative contributions of different PONs have been removed in the revised manuscript. In our view, some conclusion could be drawn from the proportions of the same organic nitrate at different sampling sites. For example, the larger proportion of oleic acid-derived organic nitrates at urban sites was consistent with the effect of cooking on SOA formation. Thus, this part was still retained in the revised manuscript.

Line 240–243, "*It is worth noting that OAKN359 also contributed to a large fraction (16–38%) of the ΣPONs and it was more abundant at urban sites (33–38%) than at rural sites (16–17%). Additionally, PSON295 and OAHN361 only accounted for small proportions of the ΣPONs. It is interesting that the proportion of PSON295 (6%) at the JN sites was markedly higher than those at other sites (<3%)*"

8. Line 234. I don't think in this paper the authors reported PNOs. Instead, the paper by He et al. 2014 and Lin et al. 2012 reported PSON in south China.

**Response:** the references have been updated in the revised manuscript by removing the reference of Ding et al. 2016 and adding the references of Lin et al 2012.

Line 244–246, "*The large proportion of monoterpene-derived organic nitrates in PONs was also observed in southeastern United States (23–44% in mole) and in East Asia (Ayres et al., 2015; Lin et al., 2012) and was ascribed to the high emissions of monoterpenes together with the relatively high production yields of PONs.*"

9. Line 242-245. These two arguments are not well supported. Since surrogate standard was used, the real concentration of acid-derived organic nitrates is hard to know. Though the oleic acid-derived organic nitrates were quantified with a considerable level, it is hard to estimate the contribution of cooking and oil processing on the SOA production from this single compound. Then you can not state like "significant influence". For the second argument on PSON295, there is no discussion on the production and loss at all. Thus, no conclusion can be drawn.

**Response:** thanks for the comment. It will be appropriate to compare the relative concentrations of oleic acid-derived organic nitrate in urban and rural area. The conclusion has been revised as follows.

Line 253–255, "*The identification and quantification of oleic acid-derived organic nitrates in this study confirm that the impact of cooking and oil processing on SOA in urban areas is more obvious than that in rural areas in eastern China.*"

The description on PSON295 here, "As for PSON295, the relatively high contribution at urban Jinan is possibly associated with the massive coal-fired industries distributed throughout the city" has been removed, and the influence on PONs concentrations from coal combustion has been discussed in Section

10. Line 250-256. The atmospheric boundary layer height could also play an important role in the diurnal trend, especially for those that nighttime concentrations were higher.

**Response:** to verify whether the boundary layer height change affected the diurnal patterns of PONs, the daytime and nighttime ratios of PONs to $PM_{2.5}$ are compared (see Fig. S3 in the supporting information). As shown, the diurnal difference in the ratios of PONs to $PM_{2.5}$ are quite similar to those in the concentrations of PONs, confirming that the diurnal variations of PONs observed in this study are not caused by the boundary layer height change.

[Figure]

[Figure]

**Figure S3.** The differences in PONs/PM$_{2.5}$ ratios during daytime and nighttime at different sampling sites.

Line 257–263, "*Figure 3 shows the average concentrations of individual PONs during daytime and nighttime at four sampling sites. Generally, the daytime concentrations of PONs were comparable to those at nighttime (p > 0.1 for most sites). However, some differences were found between the daytime and nighttime concentrations of individual PONs. Particularly, at the DY site in summer, MHN215 and PKN229 exhibited obviously higher contributions at daytime than those at nighttime (p < 0.01). To verify whether the diurnal difference was aroused by the change in boundary layer height, the PONs/PM$_{2.5}$ ratios were also compared (see Fig. S3). The very similar diurnal trends between PON concentrations and PONs/PM$_{2.5}$ ratios suggest that the boundary layer change had little influence on the diurnal difference in the concentrations of PONs.*"

11. Line 257-262. When discussing the seasonal trend, production due to emissions and reaction rate change can be important. However, the loss of PNOs due to photolysis and hydrolysis (Nah et al., 2016; Takeuchi and Ng, 2019) can be also different. These should also be considered.

**Response:** thanks for the comment. We assume that the "seasonal trend" mentioned by the reviewer here meant diurnal difference. Photolysis and hydrolysis have been considered when discussing the seasonal trends in the revised manuscript. Photolysis in the condition of intensive sunlight will reduce the PON concentrations during daytime in summer, while hydrolysis in the condition of high humidity will cause a decrease in the concentrations of PONs particularly at nighttime. Takeuchi and Ng (2019) comprehensively evaluated the hydrolysis processes of particulate organic nitrates which were generated from the oxidation of α-pinene and β-pinene and the results showed that the hydrolysis lifetime was fast, within 30 min. However, in this study, the concentrations of PONs and their ratios to PM$_{2.5}$ in moist conditions in summer were even higher than those in other seasons. With consideration of the moderate humidity in northern China, hydrolysis is not considered as the dominant factor causing the diurnal difference at the DY site in summer. In addition, the concentrations of PONs and their ratios to PM$_{2.5}$ in

summer with intensive solar radiation were not lower than those in other seasons. Furthermore, the particulate organic nitrates produced from different precursors have different photochemical aging behaviors and the photolysis rates of the organic nitrates are relatively small when compared to their production rates (Nah et al., 2016; Luke et al., 1989), so in our view photolysis is not the major factor that governed the diurnal difference of PONs in this study. The following discussion has been added in the revised manuscript.

Line 288–298, "*On the basis of the observation data, it was difficult to see any apparent effects of photolysis and hydrolysis on the concentrations and variations of PONs from temperature and humidity. Takeuchi and Ng (2019) comprehensively evaluated the hydrolysis processes of particulate organic nitrates which were generated from the oxidation of α-pinene and β-pinene and the results showed that the hydrolysis lifetime was fast, within 30 min. However, in this study, the concentrations of PONs and their ratios to PM$_{2.5}$ in the moist conditions in summer were even higher than those in other season. With consideration of the moderate humidity in northern China, hydrolysis was not considered as the dominant factor that caused the lower concentrations of PONs at nighttime than daytime at the DY site in summer. In addition, the concentrations of PONs and their ratios to PM$_{2.5}$ in summer with intensive solar radiation were not lower than those in other seasons. Furthermore, the organic nitrates produced from different precursors have different photochemical aging behaviors and the photolysis rates of the organic nitrates are relatively small when compared to their production rates (Nah et al., 2016; Luke et al., 1989), so photolysis was not the major factor that governed the seasonal differences of PONs in this study.*"

12. Line 289-290. Back trajectory analysis for this day and other sampling days showing different transportation is very helpful to support this argument.

**Response:** the back trajectory analysis has been added in the supporting information (Fig. S6).

Line 326–328, "The relatively high levels of SO$_2$ on 22 and 23 October were ascribed to the transport of plumes from coal-fired industries such as steel plants and thermal power plants in the northeast sector of Jinan City under the influence of northeast winds (see Fig. S6)."

[Figure]

**Figure S6.** Air masses clusters classified by the 24-h backward trajectories at the JN site (a) on 22 and 23 October and (b) from 25 to 31 October.

13.   Line 297-300. While this study focused on the secondary produced PNOs in the atmosphere, it is hard to believe that the NO and NO2 data has never been shown. For the discussion here, figures showing poor correlations between NOx and PONs are of equal importance as figures showing good correlation between PNOs and SO2.

**Response:** the $NO_2$ data are shown in Table 3. The NO data are only available at the DY site in summer and thus are not shown in Table 3. The scatter plot showing the correlation between $NO_2$ and PONs has been added in Figure 6 (see Fig. 6b).

[Figure]

**Figure 6b.** Correlation plots between PONs and $NO_2$ at the DY site in winter and at the JN site.

14. Line 312-314. If this is the contribution of the CI+SO2 pathway, I will double that this can affect LDK247 production significantly. 1) in this calculation, lots of parameters are needed, such as NO3 radical concentration, NO mixing ration, and SO2 mixing ratio. The values being used in this calculation should be clarified, especially for NO which is never shown elsewhere. 2) to show the influence the SO2, the author may do the calculation in an alternative way. That is to vary the SO2 concentration in a reasonable range while fixing other parameters. Then the authors may see how can this pathway contribute to the LSK247 production.

**Response:** Thanks for the comment and suggestion. The data used in the calculation has been clarified. The calculation method has been added in the revised manuscript.

Line 354–358, "*Here, the observed $SO_2$ concentration at JN site at the daytime (4.9 ppb) and night (4.1 ppb) was used. The NO concentration was assumed to be 5 ppb at daytime and 15 ppb at nighttime, respectively, according to the filed measurements at the same site in late summer/early autumn in 2014 (Wang et al., 2017). The concentration of $NO_3$ radical was calculated as 0.26 ppt at nighttime using $[NO_3] = [N_2O_5]/(K_{eq}[NO_2])$ (Sander et al. 2006), in which$[N_2O_5]=50$ ppt and $[NO_2]=35$ ppb (Wang et al., 2017).*"

Line 358–360, "*Under the conditions of fixed concentrations of the above species and only the chemical generation being considered, when the $SO_2$ concentration changed from 0.8 ppb to 8.2 ppb at the JN site, the increase in the contribution of the CIs+$SO_2$ pathway to the HDCN247 production varied from 0.5% to 15.4%.*"

15. Line 340-341. This is contradictory to the argument by the author that CI+SO2 have potential influence on PNOs formation. As elevated O3 concentration would promote CI production.

**Response:** high levels of ozone can lead to the high concentrations of OH radicals during the daytime, especially in the summer when the solar radiation is intensive. The increase in the OH radical concentration will enhance the formation of organic nitrates. In addition, the elevated concentrations of ozone also promote the production of CIs. The produced CIs are ready to react with not only $SO_2$ but also NO, $NO_2$, and CO to form some intermediate products (e.g., 3-isopropenyl-6-oxoheptanal) and finally produce organic nitrates. The contribution of the CIs+$SO_2$ pathway firstly depends on the $SO_2$ concentrations of, and secondly relies on the ozone concentrations. At the rural area of DY, during the heating period in winter, a large amount of coal was burned nearby for residential heating and cooking. However, there were rare coal combustion activities near the sampling site, because there was no heating demand. Therefore, CIs+$SO_2$ pathway did not play an important role in the formation of PONs in summer at the DY site..

Line 385–386, "*Exposed to ozone-rich atmosphere, PONs concentrations were moderately correlated*

*with ambient ozone concentrations (Fig. 8, r = 0. 55, p < 0.05).*"

16. As I mentioned before, figures showing good correlation and poor correlation are of equal importance. This comparison provides us a more clear impression of the results.

**Response:** the scatter plots showing the correlation between PONs and $NO_2$*$O_3$ at other sites have been added in Fig. 9b.

[Figure]

**Figure 9b.** Scatter plot and the correlations of PONs the concentration product of $O_3$ and $NO_2$ at nighttime at the four sites.

17. Line 374-375. Please provide NO data to support this. Since NO is one of the key species in PNOs formation, I suggest the authors to put NO, NO2 data in table one.

**Response:** the concentrations of $NO_2$ are shown in Table 3. However, the NO data are not available at present at most sampling sites, so they are not included in Table 3.

18.

Errors

1. Line 49, replace "higher" with "low"

**Response:** Thanks for the correction. It has been corrected in the revised manuscript.

2. Line 50. Change "a mechanism catalyzed by aerosol acidity" to "acid-catalyzed reactions"

**Response:** changed.

3. Line 114. Change "contained" to "shown"

**Response:** changed.

4. Lie 167. Add "with" after "identified"

**Response:** added.

5. Line 304. Delete the space before PNOs

   **Response:** deleted.

6. Figure 5, "LDKN248" should be "LDKN247"

   **Response:** changed with the new name of HDCN247.

7. Table 2, "(MW = 295, PSON 295" should be "(MW = 295, PSON 295)"

   **Response:** changed.

**References:**

Claflin, M. S. and Ziemann, P. J.: Identification and Quantitation of Aerosol Products of the Reaction of β-Pinene with NO3 Radicals and Implications for Gas- and Particle- Phase Reaction Mechanisms, The Journal of Physical Chemistry A, 122, 3640-3652, 2018.

Lin, P., Yu, J. Z., Engling, G., and Kalberer, M.: Organosulfates in Humic-like Substance Fraction Isolated from Aerosols at Seven Locations in East Asia: A Study by Ultra-High-Resolution Mass Spectrometry, Environ. Sci. Technol., 46, 13118- 13127, 2012.

Luke, W. T., Dickerson, R. R., and Nunnermacker, L. J.: Direct measurements of the photolysis rate coefficients and Henry's law constants of several alkyl nitrates, J. Geophys. Res. Atmos., 94, 14905-14921, 10.1029/JD094iD12p14905, 1989.

Nah, T., Sanchez, J., Boyd, C. M., and Ng, N. L.: Photochemical Aging of α-pinene and β-pinene Secondary Organic Aerosol formed from Nitrate Radical Oxidation, Environ. Sci. Technol., 50, 222-231, 2016.

Shi, X., Qiu, X., Cheng, Z., Chen, Q., Rudich, Y., and Zhu, T.: Isomeric Identification of Particle-Phase Organic Nitrates through Gas Chromatography and Time-of-Flight Mass Spectrometry Coupled with an Electron Capture Negative Ionization Source, Environ. Sci. Technol., 10.1021/acs.est.9b05818, 2020.

Takeuchi, M. and Ng, N. L.: Chemical Composition and Hydrolysis of Organic Nitrate Aerosol Formed from Hydroxyl and Nitrate Radical Oxidation of α-pinene and β- pinene, Atmos. Chem. Phys. Discuss., 2019, 1-39, 2019.

Kristensen, K., and Glasius, M.: Organosulfates and oxidation products from biogenic hydrocarbons in fine aerosols from a forest in North West Europe during spring, Atmospheric Environment, 45, 4546-4556, https://doi.org/10.1016/j.atmosenv.2011.05.063, 2011.